# Sex differences in the genetic regulation of the human plasma proteome

**Mine Koprulu** [1,2], **Eleanor Wheeler**[2], **Nicola D. Kerrison** [2], **Spiros Denaxas**[3,4,5,6], **Julia Carrasco-Zanini** [1,2], **Chloe M. Orkin** [7,8], **Harry Hemingway** [3,4,6], **Nicholas J. Wareham** [2], **Maik Pietzner** [1,2,9] & **Claudia Langenberg** [1,2,9] ✉

Mechanisms underlying sex differences in the development and prognosis of many diseases remain largely elusive. Here, we systematically investigated sex differences in the genetic regulation of plasma proteome (>5800 protein targets) across two cohorts (30,307 females; 26,058 males). Plasma levels of two-thirds of protein targets differ significantly by sex. In contrast, genetic effects on protein targets are remarkably similar across sexes, with only 103 sex-differential protein quantitative loci (sd-pQTLs; for 2.9% and 0.3% of protein targets from antibody- and aptamer-based platforms, respectively). A third of those show evidence of sexual discordance, i.e., effects observed in one sex only (n = 30) or opposite effect directions (n = 1 for CDH15). Phenome-wide analyses of 365 outcomes in UK Biobank did not provide evidence that the identified sd-pQTLs accounted for sex-differential disease risk. Our results demonstrate similarities in the genetic regulation of protein levels by sex with important implications for genetically-guided drug target discovery and validation.

Many aspects of human development and health, including the age of onset, prevalence, and severity of many diseases differ between sexes[1–6], but the underlying mechanisms or extent to which genetic factors contribute to any differences remain largely unknown[7–9].

The recent advancements and dropping costs of omics technologies have now made it feasible to apply them to large scale studies, providing a previously unprecedented molecular view into states of health and disease. Previous efforts have investigated the extent of sex-differences in relation to gene expression quantitative trait loci (eQTLs)[10] through a sex-stratified approach. However, the sex differential genetic regulation of the proteome has been limited to ad *hoc* investigations of protein quantitative trait loci identified in sex-combined analysis[11,12], where sex-differential or sex-discordant effects might be masked, and systematic efforts are lacking. Understanding

genetically driven sex-differences at the molecular level, specifically proteins as the biologically active entity between the genome and the phenome, is important for basic and translational genetic research, including genetically anchored drug target discovery and validation.

The large sample size of the Fenland study (sex combined sample size = 8348) and UK Biobank (sex combined sample size = 48,017) together with broad proteomic coverage across two technologies enabled systematic investigation of sex differences in the genetic regulation of plasma proteins. We contrast sex-differential protein abundance with sex-specific genetic regulation for 4775 unique proteins, targeted by 4979 unique aptamers (measured with SomaLogic) in 4403 females and 3945 males (aged 29–64) from the Fenland study[11] and 2923 unique proteins, targeted by 2923 unique antibody assays (measured with Olink) among 25,904 females and 22,113 males (aged 49–60) from

[1]Precision Healthcare University Research Institute, Queen Mary University of London, London, UK. [2]MRC Epidemiology Unit, University of Cambridge School of Clinical Medicine, Institute of Metabolic Science, Cambridge, UK. [3]Institute of Health Informatics, University College London, London, UK. [4]Health Data Research UK, London, UK. [5]British Heart Foundation Data Science Centre, London, UK. [6]National Institute of Health Research University College London Hospitals Biomedical Research Centre, London, UK. [7]Blizard Institute and SHARE Collaborative, Queen Mary University of London, London, UK. [8]Department of Infection and Immunity, Barts Health NHS Trust, London, UK. [9]Computational Medicine, Berlin Institute of Health at Charité-Universitätsmedizin Berlin, Berlin, Germany. ✉e-mail: claudia.langenberg@qmul.ac.uk

UK Biobank[13] (Supplementary Data 1) with 1838 proteins being targeted by both platforms by 1991 unique protein combinations.

We defined 'female' and 'male' sex by matching the recorded sex and sex chromosomes (XX for females and XY for males) for both studies. The recorded sex contained a mixture of self-reported sex and sex through medical records, and it was not possible to distinguish sex from gender. We acknowledge the importance of distinguishing between sex and gender in research and that chromosomal make-up does not always align with self-identified gender. For the present study, we therefore define 'sex' as the chromosomal make up of participants and do not make any inference about gender and such should also not be made from the results of our study.

## Results

### Substantial differences between the female and male plasma proteome

Most protein targets ($n = 4025$ proteins targets out of 5823 included in this study, 69.1%) showed significant sex differences ($p_{het} < 1.01 \times 10^{-5}$ for aptamer-based and $p_{het} < 1.71 \times 10^{-5}$ for antibody-based platform, respectively; see Methods) in their plasma abundance in at least one cohort, including 768 (41.7%) overlapping targets with directionally concordant effects between sexes (Fig. 1, Supplementary Data 2 and Supplementary Fig. 1). Results exemplified large differences between the sexes, with a slightly larger number of protein targets showing higher levels in males compared to females across both technologies (62.1% [$n = 1283$] and 63.8% [$n = 1851$] for antibody- and aptamer-based technologies, respectively; Fig. 1 and Supplementary Data 2). Adjustment for hormone replacement therapy/oral contraception or known sex-differential participant characteristics such as body mass index, low-density lipoprotein cholesterol (LDL) levels, alanine transaminase (ALT) levels, smoking status and the frequency of alcohol consumption attenuated only a moderate number of significant differences (15.3% [$n = 616$] for hormone replacement therapy/oral contraception and 20.5% [$n = 827$] for known sex-differential characteristics listed above, respectively; Supplementary Data 2). Proteins with the largest differences reflected sex-specific biology, e.g., specific expression in female- or male-specific tissues, such as prostate-specific antigen[14] (beta [95% confidence interval (CI)] = 1.56 [1.53–1.58], $p = 2.31 \times 10^{-2823}$, UniProt: P07288), prokineticin 1 (beta [95% CI] = 1.25 [1.24–1.26], $p = 1.97 \times 10^{-7019}$, UniProt: P58294), or follicle stimulating hormone (beta [95% CI] = −1.22 [−1.21 to −1.23], $p = 1.7 \times 10^{-6862}$, UniProt: P01215), while some others likely reflected the effect of sex-differences in body composition on plasma abundance of specific protein targets, such as leptin[15,16] or adiponectin[16–18]. We also observed strong sex-differences in established cardiovascular diagnostic markers such as NT-proBNP (beta [95% CI] = −0.78 [−0.74 to −0.82], $p = 3.33 \times 10^{-338}$, UniProt: P16860) and troponin T (beta [95% CI] = 0.83 [0.79–0.87], $p = 9.86 \times 10^{-388}$, UniProt: P45379).

Males and females do not only differ by disease onset and severity, but also in drug response and a higher frequency of adverse drug reactions is observed in females compared to males[19]. We identified a total of 129 proteins that are the targets of already approved drugs or drugs in early clinical trials[20] and showed significant differences between sexes in plasma abundance that were directionally consistent across cohorts. For example, fibrinolytic agents, such as Tenecteplase or Urokinase, that target plasmin (beta [95% CI] = −0.21 [−0.19 to −0.23], $p = 3.83 \times 10^{-94}$, UniProt: P00747) have been described to be differentially effective in female and male patients in post stroke therapy[21]. Although plasma protein levels are not the primary target for most of those drugs, our results can potentially help understanding sex-differential drug effects.

### Genetic regulation of plasma proteins is largely comparable across the sexes

We next performed sex-stratified genome-proteome-wide association studies to systematically identify sex-differential protein quantitative trait loci—'sd-pQTLs' (Supplementary Data 3 and 4), defined as statistically significant differences in the association of the variant with the abundance of a given protein target between the sexes. We identified a large number of pQTLs ($p < 5 \times 10^{-8}$) in each sex ($n_{females} = 7424$, $n_{males} = 6546$ pQTLs aptamer-based and $n_{females} = 18,307$, $n_{males} = 14,305$ pQTLs for antibody-based technology; Supplementary Fig. 2). Importantly, we observed that around 15% of pQTLs identified in females ($n = 1149/7424$) or males ($n = 976/6546$) for aptamer-based technology and around 7% of pQTLs identified in females ($n = 1332/18307$) or males ($n = 995/14305$) for antibody-based technology were not significant ($p < 5 \times 10^{-8}$) in the sex-combined analyses, as their effect were likely masked in the sex-combined analyses. Despite the large number of pQTLs identified in each sex, only very few pQTLs showed significant differences in effects between males and females (i.e., sex-differential effects), with 15 ($p_{het} < 1.01 \times 10^{-11}$) and 88 ($p_{het} < 1.71 \times 10^{-11}$) sd-pQTLs being identified for aptamer and antibody-based platforms, respectively (Supplementary Data 3 and 4).

The sd-pQTLs fell into three broad categories: (i) 72 sd-pQTLs were significant in both sexes with the same direction of effect yet differing magnitudes, (ii) 30 sd-pQTLs were only significant in one sex, and (iii) one sd-pQTL was significant in both sexes but with opposite effect directions. We refer to the latter two categories as 'sex-discordant' but acknowledge that sd-pQTLs of this category might reach significance in the opposite sex in yet larger studies while still being characterized by substantial effect size differences. In general, identified examples were predominantly ($n = 72$; 69.9%) sex differential rather than sex-discordant (i.e., only evident in one sex or different effect directions between sexes). In addition, most sd-pQTLs reside close to the cognate gene (cis-pQTLs; $n = 72$ across technologies, 69.9%).

### Strong sex-differential pQTLs have roles in reproduction but also beyond

We observed no enrichment of sd-pQTLs on the X-chromosome or among druggable targets ($p > 0.05$). We did not observe a clear bias towards protein encoding gene expression explicitly in reproductive tissues or breast for the proteins for which at least one cis or trans sd-pQTL was identified[10,22]. Overall, 31 sd-pQTLs showed sex-discordant effects, with strong evidence of an effect in one but not the other sex ($p > 5 \times 10^{-8}$) for all, except for cadherin-15 (CDH15) where the cis sd-pQTL (rs113693994) was significant in both males and females yet showed opposite effect directions. Some of the sex-discordant pQTLs mapped to proteins with established roles in only one of the sexes. For example, we identified cis sd-pQTLs for pregnancy zone protein (PZP) that replicated across both technologies. The cis-sd-pQTL for PZP was significant only in females in the aptamer-based technology and with an almost three times higher effect size in females in the antibody-based technology. (Fig. 2 and Supplementary Data 3 and 4). Likewise, two protein targets with sd-pQTLs that were significant in males only (prostate and testis expressed protein 4 [PATE4, rs499684] and Kunitz-type protease inhibitor 3 [SPIT3, rs6032259]) have been reported to be involved in male fertility (Fig. 2 and Supplementary Data 3). PATE4 has a reported function as a factor contributing to the copulatory plug formation in male fecundity in mouse models[23] and is predominantly expressed in prostate and testis[24]. Similarly, SPIT3, encoded by SPINT3, is reported to be predominantly expressed in epididymis although the mouse orthologue of this gene was reported to be dispensable for fertility[25,26]. Although their sex-specific biological function was not clear, we also identified sd-pQTLs with consistent effect direction for neural cell adhesion molecule 1 (NCAM-1), oxytocin-neurophysin 1 (NEU1) and ectonucleotide pyrophosphatase/phosphodiesterase family member 7 (ENPP7) that replicated across both platforms (Fig. 2 and Supplementary Data 3 and 4).

For six proteins, insulin-like 3 (INSL3), acrosomal vesicle protein 1 (ACRV1), tetraspanin 8 (TSPAN8), apolipoprotein E (APOE),

carboxypeptidase E (CPE) and CDH15, we identified more than one sd-pQTL, further supporting a sex-differential or −discordant genetic regulation (Supplementary Data 4) for these proteins. While sex-differential effects for INSL3 and ACRV1 might be explained by almost exclusive expression in male tissues, like testis, effects for the other three examples are less clear and might even differ by locus. For example, the cis-sd-pQTL rs429358>C for APOE encodes the ε4-allele associated with higher risk of late-onset Alzheimer's disease and the stronger effect in females is in line with a higher prevalence of Alzheimer's disease among them[27]. In contrast, the male-specific effect of the trans-sd-pQTL for APOE (rs964184) maps to a region on chromosome 11 strongly associated with lipid metabolism and harbouring multiple apolipoproteins, more likely reflecting its role as a carrier for lipoproteins.

We observed two trans sd-pQTLs for INSL3 both of which were only significant in males (Supplementary Data 4). INSL3 is a small peptide from the insulin-like hormone superfamily. In the human foetus, INSL3 is produced by foetal Leydig cells after gonadal sex determination around weeks 7 to 8 post coitum and has an important role in testis descent. In line with its foetal role, damaging variants in *INSL3* have been reported to cause autosomal dominant cryptorchidism[28]. INSL3 is also produced by the ovarian follicular theca cells in females which is known as the Leydig cell counterpart in the females, with the INSL3 knockout mouse displaying phenotypes which

suggest a role in the number of healthy growing follicles[29,30]. In adults, its expression was reported to peak around early adulthood and show a gradual decrease throughout life afterwards whereas in females the levels are reported to be impacted diseases which impact the number of growing follicles such as polycystic ovary syndrome or menopause[29,30]. In line with the reported functions, INSL3 is reported to be expressed only in testis and ovary with a much lower level of expression in ovary compared to testis[24].

For ACVR1C, we observed two sex-discordant pQTLs where the cis-sd-pQTL for ACRV1 was only significant in females and trans-sd-pQTL was significant in only males, potentially with reverse mechanisms of impact (i.e., one pQTL acting to decrease whereas the other acting to increase the levels of ACRV1). ACRV1 is exclusively expressed in testis[24] and has a function during spermatogenesis[31].

We identified three sd-pQTLs for TSPAN8, all of which were significant in both sexes. The cis sd-pQTL was stronger in males compared to females whereas the two trans sd-pQTLs were stronger in females. TSPAN8 belongs to transmembrane 4 superfamily and has been associated with different carcinomas[32–35]. Although there were no direct links between TSPAN8 and sex-specific pathways, interestingly, SRY-Box Transcription Factor 9 (SOX9), which plays an important role in sex determination, was identified as a key transcriptional regulator of TSPAN8 in metastasis by pancreatic ductal adenocarcinoma[36].

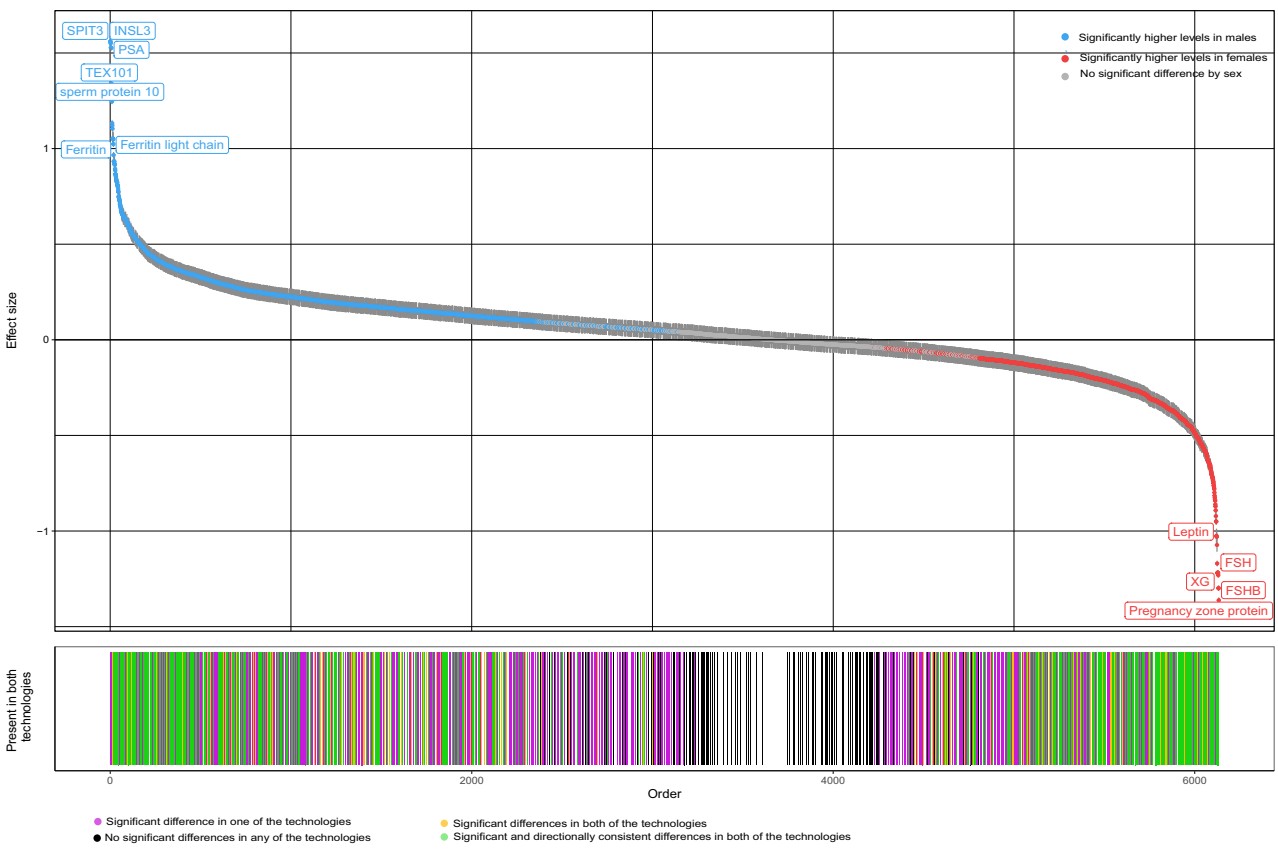

**Fig. 1 | Sex differences in the abundance of 5823 unique proteins measured by 4979 unique aptamers and 2923 unique antibody assays.** Linear regression models were used to test the association of sex with the protein abundance in each cohort. The protein targets were ordered by their effect size in males. *Top panel*: The top panel shows the proteins for which the plasma abundance significantly differed by sex in at least one technology ($p_{het} < 1.01 \times 10^{-5}$ for aptamer-based and $p_{het} < 1.71 \times 10^{-5}$ for antibody-based technology were used as Bonferroni-corrected thresholds respectively). The proteins were coloured blue if they had significantly higher levels in males and red if they had higher levels in females. If the protein target was significant in both of the technologies, the effect size estimate from the more significant study was displayed. The dark grey vertical lines represent the 95% confidence intervals for the effect size estimates. *Bottom panel*: The bars in the bottom panel represent the proteins which were targeted by both aptamer-based and antibody-based platforms. The lines were coloured lighter green if the finding was significant and directionally consistent in both technologies, yellow if the finding was significant but not directionally consistent across technologies, lilac if the finding was only significant in one of the technologies and black if the finding was not significant in any of the technologies. Results can be found in Supplementary Data 2.

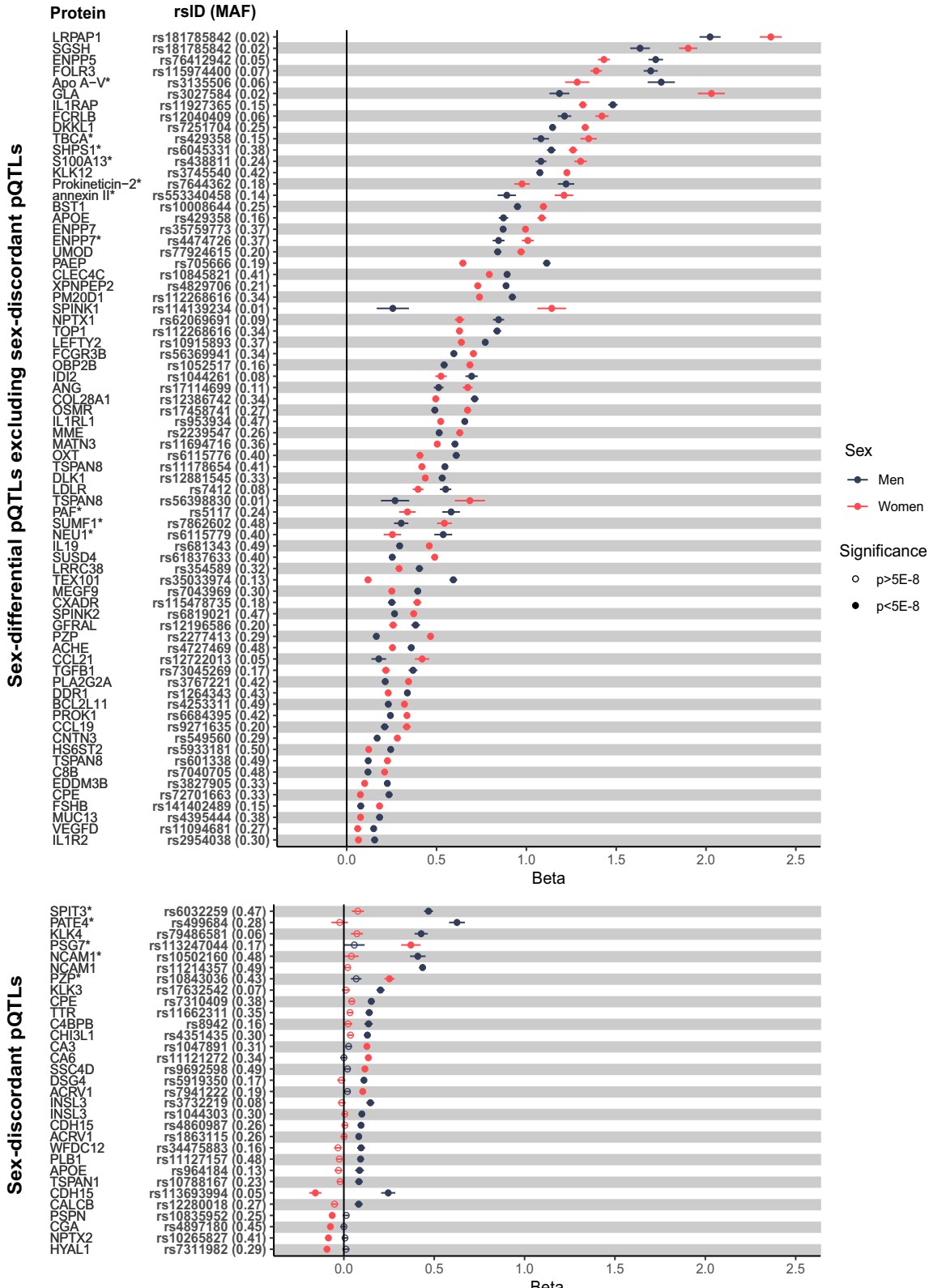

**Fig. 2 | Forest plot of all identified sex-differential protein quantitative trait loci (sd-pQTLs) from both aptamer-** ($p_{het} < 1.01 \times 10^{-11}$) **and antibody-based** ($p_{het} < 1.71 \times 10^{-11}$) **technologies.** The bottom panel presents sex-discordant pQTLs (not significant ($p < 5 \times 10^{-8}$) in one sex or has opposing effect directions), whereas the top panel presents the remaining sex-differential pQTLs. The significant ($p < 5 \times 10^{-8}$) pQTLs in each sex are represented by filled circles and non-significant ones are represented by hollow circles. Linear regression models were used to identify pQTLs in each sex in each cohort. Horizontal lines represent 95% confidence interval for the effect size estimate of each finding. The sample sizes and detailed summary statistics for each of the findings can be found in Supplementary Data 3 and 4. Proteins with an asterisk (*) were measured using the aptamer-based technology, otherwise using antibody-based technology. MAF minor allele frequency.

The cis-sd-pQTL (rs113693994, $\text{beta}_{\text{females}}$[95% CI] = −0.16 [−0.12 to −0.19], $p_{\text{females}} = 9.8 \times 10^{-20}$, $\text{beta}_{\text{males}}$[95% CI] = 0.25 [0.21–0.28], $p_{\text{males}} = 9.32 \times 10^{-35}$) for CDH15 was the only example observed in this study where a pQTL was significant in both sexes but with opposite effect directions. It is therefore a pQTL that has not been identified in a sex-combined study ($\text{beta}_{\text{sex\_combined}}$[95% CI] = 0.02 [−0.01–0.04], $p_{\text{sex\_combined}} = 0.23$). The sex-differential genetic regulation for CDH15 plasma levels was further supported by a male-specific trans-pQTL (rs4860987). CDH15 acts a cell adhesion molecule that is involved in facilitating cell-cell adhesion and preserving tissue integrity and is highly expressed in brain and muscle[24]. While a sex-differential regulation at the cis-locus remains elusive, the trans-sd-pQTL (rs4860987) maps into a region harbouring several uridine diphosphoglucuronosyltransferases involved in the clearance of, among others, steroid hormones and has further been reported to associate with sex-hormone binding globulin levels[37] possibly suggesting a sex-hormone dependency of CDH15 plasma level regulation. Similarly, CPE which acts as an exopeptidase essential for the activation of peptide hormones (e.g., insulin) and neurotransmitters had both a cis and a trans sd-pQTL with both sd-pQTLs having stronger effects in males compared to females (Supplementary Data 4). CPE has been implicated to have a role in osteoclast differentiation. *Cpe* knockout mice displayed low bone mineral diversity, increased osteoclastic activity as well as being obese and displaying a diabetic phenotype[38,39]. However, neither CDH15 nor CPE have a clearly established sex-specific function or disease associations to date, although the fact that these two proteins have both cis and trans sex-specific genetic regulation might suggest their potential involvement in a sex-specific biological function.

### Phenotypic follow-up of sd-pQTLs

We next conducted phenome-wide association analyses (PheWAS) to identify potential sex-differential phenotypic consequences of sd-pQTLs across 365 diseases with more than 2500 cases in UK Biobank. We identified 82 unique significant variant-outcome associations (Bonferroni corrected significance threshold of $p < 1.59 \times 10^{-6}$ and $p < 9.13 \times 10^{-6}$ for antibody- and aptamer-based technologies, respectively, corrected for the number of unique variants and disease outcomes) between sd-pQTLs and disease risk in at least one sex. Despite numerous significant associations of sd-pQTLs with disease outcomes in at least one sex (Fig. 3), none of the associations passed our stringent multiple testing threshold to declare significance for heterogeneity between sexes (Supplementary Data 5 and 6) (Bonferroni corrected significance threshold of $p_{\text{het}} < 1.59 \times 10^{-6}$ and $p_{\text{het}} < 9.13 \times 10^{-6}$ for antibody- and aptamer-based technologies, respectively, corrected for the number of unique variants and disease outcomes). In general, this leaves the downstream physiological or pathological consequences of the identified sd-pQTLs yet to be determined (Fig. 3). We note, however, that we observed nominal significant support ($p_{\text{het}} = 1 \times 10^{-3}$) for the cis-sd-pQTL for APOE on a sex-differential risk on dementias, replicating previous findings of a higher risk among women carrying the ε4-allele[40] (Supplementary Data 6).

### Discussion

Biological sex is an important, yet historically neglected modifier of disease risk and progression. Our knowledge about the mechanisms through which the sex differences act remains relatively limited, with the majority of research focussing on the effects of sex hormones or proteins encoded on the X-chromosome. While these are important factors that account for some of the differences between the sexes, there is a need to systemically better understand differences that potentially translate into sex-differential disease risk.

In this study, we observed substantial variation between the female and male plasma proteome, including over 4000 proteins targets being differentially expressed between sexes. We demonstrate

that only few (<3%) of those show evidence of sex-differential regulation through germline genetic variants. Our study highlights two important conclusions. Firstly, the fact that we observe sd-pQTLs for a very small percentage of protein targets despite large differences in plasma protein levels emphasizes that other intrinsic (e.g., hormone profiles) and extrinsic mechanisms (e.g., sex-differential lifestyle and risk-factor profiles) are also likely to strongly influence the observed sex differences. This finding is in line with what has been reported for sex-differences observed for complex diseases, sex-differential genetic loci being identified for only a small proportion of common diseases[8]. Secondly, our results suggest that the use of pQTLs in biomedical research, specifically for drug target discovery and causal inference will—with few exceptions—likely generate findings that are generalisable across sexes for the studied protein targets. However, future studies should continue to evaluate sex differences as increased power could or broader proteomic coverage potentially uncover additional examples with biologically relevant sex-discordant effects.

Our findings are in line with previous sex-stratified analyses of tissue-specific gene-expression[10], including the observation that most sd-pQTLs act in a sex-differential rather than sex-discordant manner. It has been recently demonstrated that trait variance difference between sexes can predominantly be explained by sex-differential 'amplification effect' as an aggregated effect across the genome (i.e., same effect direction yet different magnitudes of strength between sexes)[41], which might be one explanation for the very few locus-specific effects we observed here. Alternatively, there might also be higher order interactions of genomic loci with sex hormones that determine gene expression.

Although only few, we did identify some sex-discordant genetic effects, with some reflecting sex-specific biology (e.g., PATE4, SPIT3, PZP) that might be acting via steroid hormone responsive elements. Some of the other effects might possibly be the result from differential environmental exposures between the sexes, as suggested previously for genetic variants affecting the risk for gout that may act through differential alcohol consumption[42,43].

The restriction to proteins measured in plasma represents a notable limitation of our study, as sex-differential proteogenomic effects within tissues may not be systematically reflected in plasma via secretion, natural cell turnover, or leakage. We obtained some evidence that larger sample sizes (as seen with higher number of sd-pQTLs identified in UK Biobank) can identify a greater number of significant sd-pQTLs, but most act as weak modifiers of strong overall effects at protein encoding loci, and larger studies may possibly reveal even more subtle differences in regulation for the previously targeted proteins. Our study was performed among middle-aged participants of European ancestry, hence, it is important to scale this study to different ancestries from across the lifespan to better understand genetic factors mediating the sex-differential proteome.

Although we aimed to account for known variables that differ by sex in our non-genetic analyses, we acknowledge that this approach might overlook potential cofounders for particular proteins that was difficult to account for in this systematic effort. Given the incomplete proteomic coverage ($n = 4775$ and $n = 2923$ unique proteins targeted by aptamer- and antibody-based platforms respectively, within a spectrum of over 20,000 proteins without taking post-translational modifications or different isoforms into account) as well as limited coverage of the genomic variant spectrum (i.e., rare variants, potentially ancestry-specific effects or detailed X-chromosome inactivation models that we were not able to investigate), future studies might uncover new sd-pQTL signals as genomic and proteomic coverage continues to improve. It is also important to note that inconsistencies between platforms could be due to a wide range of reasons including but not limited to: (i) differences in power between the different cohorts, (ii) differences in cohort-related participant and sample characteristics, (iii) technological differences between the platforms

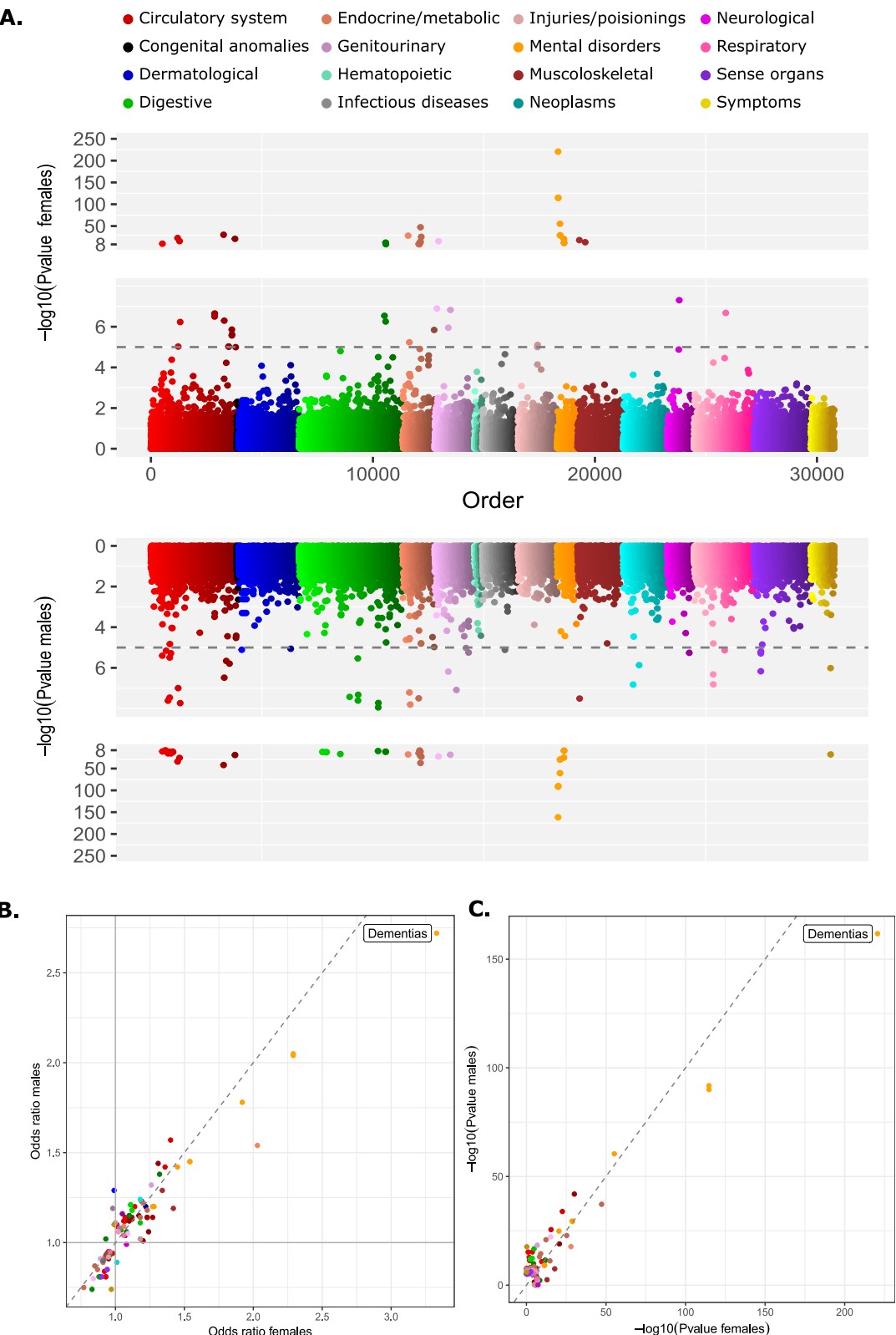

**Fig. 3 | Phenotypic follow up of the identified sd-pQTLs with 365 disease outcomes with more than 2500 cases in UK Biobank.** Logistic regression models were used to calculate associations between genetic variants and disease outcomes in each sex. **A** Miami plot of association of 100 unique variants driving 103 sex-differential pQTLs (sd-pQTLs) with 365 disease outcomes among females on the top and among males at the bottom panel. *x*-axis contains each of the sd-pQTL−disease outcome pairs, ordered by their phecodes within each disease category. The associations have been coloured by disease categories. The horizontal dashed line represents a suggestive significance threshold of ($p < 1 \times 10^{-5}$). **B** Comparison of odds ratios for the sd-pQTL−disease associations which meet the suggestive significance threshold ($p < 1 \times 10^{-5}$) in males or females. The diagonal dashed line represents the equality line ($x = y$). **C** Comparison of −$\log_{10}$ transformed *P* values for the sd-pQTL−disease associations which meet the suggestive significance threshold ($p < 1 \times 10^{-5}$) in males or females. The diagonal dashed line represents the equality line ($x = y$).

(e.g., targeting different isoforms, differing epitope effects on the assay binding).

Our study demonstrates mostly consistent genetic regulation of plasma proteins across the sexes based on two large studies with different technologies. The few exceptions could likely be explained by specific expression of protein targets in male/female tissues, potential relation to the effect of steroid hormones or differential environmental exposure. Our results that collectively add to an emerging body of literature, that strong differences in health between the sexes later in life cannot be fully explained by sex-differential or even sex-discordant effects of genetic susceptibility in individual genetic loci.

## Methods

### Study participants

The Fenland study[44] is a population-based cohort of 12,435 participants of generally white-European ancestry, born between 1950 and 1975 who underwent detailed phenotyping at the baseline visit from 2005 to 2015. Participants were recruited from general practice surgeries in the Cambridgeshire region in the UK. The participants were excluded from the study if they were (i) clinically diagnosed with diabetes mellitus or a psychotic disorder, or (ii) pregnant or lactating, (iii) unable to walk unaided, or (iv) had a terminal illness. The study was approved by the Cambridge Local Research Ethics Committee (NRES Committee – East of England, Cambridge Central, ref. 04/Q0108/19) and all participants provided written informed consent.

This study used the largest subset of individuals from the Fenland study (Supplementary Data 1). 8348 samples with both genotype information and proteomics measurements were taken forward for analyses after excluding ancestry outliers, related individuals or samples which have failed proteomics QC. The samples were well-balanced in terms of the participants from each sex: 4403 (52.7%) females and 3945 (47.3%) males were included in the study. Sex variable in Fenland study was based on general practitioners (GP) records. We only included participants with matching entries for the recorded sex and sex chromosomes (XX for females and XY for males). Individuals without matching entries were excluded from the study as a part of quality control as a mismatch can be indicative of issues with genotyping protocol.

UK Biobank is a large-scale, population-based cohort with deep genetic and phenotypic data with the full cohort consisting of approximately 500,000 participants[13]. The participants were recruited across centres in United Kingdom and were aged 40 to 69 years at the time of recruitment[13]. Ethics approval for the UK Biobank study was obtained from the North West Centre for Research Ethics Committee (11/NW/0382)[13] and all participants provided informed consent. This study used the subset of European-ancestry individuals from UK Biobank where both genotype and proteomics measurements were available after excluding ancestry outliers or samples which have failed genomic or proteomics QC ($n = 48,017$). 25,904 (53.9%) females and 22,113 (46.1%) males were included in the study (Supplementary Data 1). Sex in UK Biobank had two definitions, one was based on sex chromosomes (field 22001) and the other was contained a mixture of the sex the NHS had recorded for the participant and self-reported sex (field 31). We only included participants with matching entries for the recorded sex (from medical records or self-reported) and sex chromosomes (XX for females and XY for males) as a mismatch can be indicative of issues with genotyping protocol.

### Genotyping and imputation

The Fenland-OMICS samples have been genotyped using the Affymetrix UK Biobank Axiom array. Sample-level and variant level QC criteria were applied as described elsewhere[44]. In summary, the genotyped data was imputed to the HRC (r1) panel[45] using IMPUTE4 (https://jmarchini.org/software/) for the autosomes and Sanger Imputation Server for chromosome X (https://imputation.sanger.ac.uk/). The data was also imputed to the UK10K and 1000 Genomes Project 3 panels using and Sanger Imputation Server for both autosomes and chromosome X[46]. Additional variants gained from the UK10Kp + 1KGp3 imputation were added to the HRC imputed dataset. For basic quality control, variants were filtered for minor allele count (MAC) ≥ 3 using BCFtools[47] and INFO ≥ 0.4 using QCTOOL v2.0.2 (https://www.well.ox.ac.uk/~gav/qctool_v2/) to eliminate variants with low imputation quality.

The UK Biobank samples were genotyped using the Affymetrix UK BiLEVE or the Affymetrix UK Biobank Axiom arrays. The following QC criteria was applied to the genotyping data (a) routine quality checks carried out during the process of sample retrieval, DNA extraction, and genotype calling; (b) checks and filters for genotype batch effects, plate effects, departures from Hardy Weinberg equilibrium, sex effects, array effects, and discordance across control replicates; and (c) individual and genetic variant call rate filters as previously described[13]. Only single nucleotide polymorphisms were included in the analyses.

Genomic build GRCh37 was used throughout this study.

### Proteomic measurements

**Aptamer-based platform.** Fasting proteomic profiling of EDTA samples from Fenland study participants was performed by SomaLogic Inc. using the SOMAscan proteomic assay (v4). Relative protein abundances of 4775 human protein targets were measured by 4979 aptamers (SomaLogic V4). The quality control of the proteomic measurements has been described in detail previously[44]. Briefly, hybridization control probes were used to generate a hybridization scale factor to account for variation in hybridization within runs. A ratio between each aptamer's measured value and a reference value were computed to control for total signal differences between samples due to variation in overall protein concentration or technical factors. The median of these ratios was computed and applied to each dilution set (40%, 1% and 0.005%). Samples were removed if they were deemed by SomaLogic to have failed or did not meet our acceptance criteria of 0.25–4 for all scaling factors. In addition to passing SomaLogic QC, aptamers were filtered to only include human protein targets for subsequent analysis ($n = 4979$). Aptamers' target annotation and mapping to UniProt[48] accession numbers as well as Entrez gene identifiers[49] were provided by SomaLogic and these were used those to determine genomic positions of protein encoding genes.

**Antibody-based platform.** The UK Biobank proteomic measurements were conducted by antibody-based Olink technology, Explore 3072 platform which uses Proximity Extension Assay[50]. In summary, each protein is targeted by two unique antibodies with unique complimentary oligonucleotides, which only hybridize when they come into close proximity. This is subsequently quantified by next-generation sequencing. Normalized protein expression units, which are reported on a log2 scale, are generated by normalization of the extension control and further normalization of the plate control. Further details about antibody-based proteomic measurements and QC have been described elsewhere, including the exclusion of samples due to poor quality and selective measurements with assay warnings[51].

### Sex-differences in protein abundances

We assessed the differential abundance levels of the 4979 SomaLogic V4 aptamers between sexes in Fenland study. To estimate the effect of sex, a linear regression model was for implemented in R 3.6 for each protein target, by using the inverse rank normalized proteomic values and including covariates age and test site in the model. A stringent Bonferroni-corrected significance threshold (corrected for $n = 4979$ aptamers; $p < 1.01 \times 10^{-5}$) was applied.

2923 protein targets from Olink Explore 3072 platform in UK Biobank were inverse rank normalized and subsequently restricted cubic splines function was applied through 'rsc' function of 'Hmisc' package to regress out technical covariates such as month of the blood

draw, time that blood was drawn, fasting status and sample age in R v4.2.2. Similarly, the effect of sex in abundance levels of the 2923 protein targets from Olink Explore 3072 platform were assessed through a linear regression model in R v4.2.2, using the inverse rank normalized residuals and including covariates age, age² and proteomic batch in the model. A stringent Bonferroni-corrected significance threshold (corrected $n = 2923$ assays; $p < 1.71 \times 10^{-5}$) was applied.

Sensitivity analyses were performed by including (a) participants who have undergone hormone replacement therapy or use oral contraception, or (b) for known sex-differential participant characteristics which were body mass index (BMI), low density lipoprotein (LDL) cholesterol levels, ALT levels, smoking status and the frequency of alcohol consumption (Supplementary Data 1) and (c) for each of these factors individually (Supplementary Data 2) as additional covariates in the analyses. The continuous variables (BMI, LDL and ALT) were inverse rank normalized before being included as covariates.

To annotate druggable protein targets, we have merged the protein targets covered by aptamer- and antibody-based platforms with the list of druggable genes from Finan et al.[20]. based on common Ensembl gene IDs.

### Sex-stratified protein genome-wide association analysis (pGWASs)

For the aptamer-based platform, the protein abundances for 4979 aptamers measured in Fenland study were inverse rank normalized and regressed for the same covariates used in the proteogenomic discovery analysis of Fenland study[44], which were age, test site and the first 10 genetic principal components. The residuals for each sex were used in the subsequent association analyses.

The fastGWA software[52] was used to perform linear regression analysis through GCTA version 1.93.2 for the sex-stratified genome-wide association analyses (GWASs) in each sex for each protein target. Further variant level QC was also applied and only variants with MAC $\geq 3$, INFO $\geq 0.4$, genotype missingness rate $< 5\%$ and MAF $> 1\%$ were included in the downstream analyses.

For the antibody-based platform (i.e. the protein abundances for 2923 assays measured in UK Biobank), the same residuals from the analyses of sex-differences in protein abundances (i.e., inverse rank normalized and technical covariates regressed out) were taken forward. Sex-stratified GWASs were performed using REGENIE v.3.4.1[53] through performing two steps, as implemented by the software. In the first step, a whole-genome regression model is fitted for each phenotype to generate a covariate, which is subsequently included in the second step to allow for computationally-efficient analyses of a large number of phenotypes while also accounting for relatedness among samples. For the first step, only high-quality SNVs passing the stringent QC criteria of MAF $> 1\%$, MAC $> 100$, Hardy-Weinberg equilibrium $p$-value $< 1 \times 10^{-15}$ and genotype missingness rate $< 10\%$ were used and SNPs were pruned for linkage-disequilibrium (LD), specified for 1000 variant windows, 100 sliding windows and $r^2 < 0.8$ through Plink v.1.9. Subsequently, step 2 was applied to conduct sex-stratified GWASs for 2923 protein targets with additional per-marker QC filters of MAC $> 50$, MAF $> 1\%$ and INFO $> 0.4$. Covariates included in the proteogenomic discovery of UKBB cohort[51], age, age², proteomic batch, genotyping batch and first 10 principal components were also included as covariates in the linear regression model.

### Heterogeneity analysis to identify sex-differential pQTLs

We performed an inverse-variance fixed effects meta-analysis for each protein target using female-only and male-only summary statistics through METAL (v.2011-03-25)[52] to assess the heterogeneity in the genetic associations between sexes for each platform. Assessment of heterogeneity of results from a meta-analysis of sex-stratified results approximates individual level interaction tests, yet is several-fold more computationally and time-efficient. We additionally performed individual level interaction tests (G*S) for all identified significant sd-pQTLs identified in UK Biobank ($n = 88$) as sensitivity analyses, and observed strong correlation of the $-\log10$($p$-values) from both approaches ($r = 0.98$) with no evidence of a loss in significance.

We defined sex-differential loci as those pQTLs which were significant in at least one sex ($p < 5 \times 10^{-8}$) and showed statistically significant differences in their association between sexes. We used a proteome and genome-wide Bonferroni corrected significance threshold ($p_{het} < 1.01 \times 10^{-11}$ and $p_{het} < 1.71 \times 10^{-11}$ respectively for aptamer- and antibody-based platforms) for heterogeneity $p$-value to define sex-differential protein quantitative trait loci (i.e., sd-pQTLs). The sex-differential pQTLs which were either only significant in one sex or had opposite effect directions between sexes were further categorized as sex-discordant.

Significant genomic regions were defined by 1 Mb regions (±500 Kb on either side) around any variant with significant heterogeneity. The MHC region (chr6: 25.5–34.0 Mb) was treated as a single region. The regional sentinel variant for each genomic locus was defined as the most significant variant within the region. Variants were defined as cis-pQTLs if they were within the 1 Mb window (±500 Kb on either side) of the protein encoding gene and defined as trans-pQTLs if they were not within the 1 Mb window.

### Enrichment of sex-differential pQTLs

To assess whether the sd-pQTLs were enriched for certain characteristics (i.e., being located on X-chromosome or being a draggable target as defined by Finan et al.[20].), we conducted Chi-square tests. Additionally to assess, whether there was evidence of the target of sd-pQTLs being expressed in reproductive tissues or breast, we have used expression profiles reported by Human Protein Atlas[22].

### Phenome-wide association study (PheWAS)

We have tested whether any of the significant sd-pQTLs showed heterogeneity between sexes in terms of their disease associations across the phenome. For this purpose, in each sex, we tested the association of the unique sd-pQTL variants with 365 binary diseases with more than 2500 cases in UK Biobank. The binary disease categories were collated through clinical entities named 'phecodes' in UK Biobank, which were defined using the International Classification of Diseases, 10th Revision (ICD-10) and the International Classification of Diseases, 10th Revision, Clinical Modification (ICD-10-CM) codes from electronic health records, available in UK Biobank[54]. We tested the association of each sd-pQTL with each phecode in each sex, using a logistic regression model in R v3.6 and adjusting for age, genotype batch, test centre, and the first ten genetic principal components in unrelated European participants. We have subsequently meta-analysed the female-only and male-only summary statistics using a fixed-effects meta-analyses through *metafor* package in R v3.6 to assess the heterogeneity of the association between sexes. To correct for multiple testing, heterogeneity $p$-value threshold for PheWAS was defined as $p_{het} < 9.13 \times 10^{-6}$ and $p_{het} < 1.59 \times 10^{-6}$ for aptamer- and antibody-based platforms respectively, which were corrected for the number of unique sd-pQTL variants and number of phenotypes tested ($n = 365$) in each platform.

### Data availability

Data from the Fenland cohort can be requested by bona fide researchers for specified scientific purposes via the study website (www.mrc-epid.cam.ac.uk/research/studies/fenland/information-for-researchers/). Sex-stratified summary statistics will be made available in the GWAS Catalog upon publication. Access to the UK Biobank genomic, proteomic and phenotype data is open to all approved health researchers (http://www.ukbiobank.ac.uk/). This research has been conducted using the UK Biobank resource under the application 44448. Summary statistics for significant sex-differential pQTLs and

approximately 1 Mb (±500 Kb on either side) surrounding regions can be found here: https://doi.org/10.5281/zenodo.15061671. Sex-stratified and sex-combined genome-wide summary statistics for all protein targets in this study will be available on https://omicscience.org/ upon publication.

## Code availability

Associated code and scripts for the analyses can be found here: https://github.com/MRC-Epid/sex_specific_pGWAS.

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

## Acknowledgements

We are grateful to all Fenland volunteers and to the General Practitioners and practice staff for assistance with recruitment. We thank the Fenland Study Investigators, Fenland Study Co-ordination team and the Epide-miology Field, Data and Laboratory teams. SomaLogic proteomic mea-surements were supported and governed by a collaboration agreement between the University of Cambridge and SomaLogic. The Fenland Study (DOI 10.22025/2017.10.101.00001) is funded by the Medical Research Council (MC_UU_12015/1, C.L., N.J.W.). We further acknowl-edge support for genomics from the Medical Research Council (MC_PC_13046, C.L., N.J.W.). This work is supported by the Medical Research Council (MC_UU_00006/1 – Etiology and Mechanisms) (C.L., E.W., M.P., N.K., and N.J.W.). M.K. is supported by Gates Cambridge Trust. H.H. is supported by Health Data Research UK and the NIHR Uni-versity College London Hospitals Biomedical Research Centre. S.D. is supported by a) the BHF Data Science Centre led by HDR UK (grant SP/19/3/34678), b) BigData@Heart Consortium, funded by the Innova-tive Medicines Initiative-2 Joint Undertaking under grant agreement 116074, c) the NIHR Biomedical Research Centre at University College London Hospital NHS Trust (UCLH BRC), d) a BHF Accelerator Award (AA/18/6/24223), e) the CVD-COVID-UK/COVID-IMPACT consortium and f) the Multimorbidity Mechanism and Therapeutic Research Collabora-tive (MMTRC, grant number MR/V033867/1). J.C.Z. was supported by a 4-year Wellcome Trust PhD Studentship and the Cambridge Trust.

## Author contributions

C.L. conceptualised and supervised the project. M.K., M.P. and C.L. designed the analyses and drafted the manuscript. N.J.W. is PI of the Fenland study and C.L. and N.J.W. generated proteomic measurements in Fenland through a collaborative agreement with Somalogic. M.K., E.W., S.D., N.K., J.C.Z., H.H. and M.P. performed quality control, data preparation, or statistical and bioinformatic analyses. S.D. and H.H. developed the phenotyping algorithms to define disease outcomes. C.M.O. contributed to defining sex in this study and provided insights into broader concepts of sex and gender in research. All authors con-tributed to the interpretation of the results and critically reviewed the manuscript.

## Competing interests

E.W. is now an employee of AstraZeneca. The remaining authors declare no competing interests.
