## [Transparent Peer Review file · Nature Communications]

Sex differences in the genetic regulation of the human plasma proteome

Corresponding Author: Professor Claudia Langenberg

Version 0:

Reviewer comments:

Reviewer #1

(Remarks to the Author)

In this manuscript, Koprulu et al. evaluate the presence of sex-specific genetic associations for circulating protein levels. This is an extremely important aspect of the contemporary approach to mapping genetic determinants of human phenotypes. It is also a fundamental aspect in the process of understanding the nature of gene expression regulation, as well as whether differences between sexes are at play and how to take them into account when conducting genetic association studies. The authors should be commended for this work.

Their work is timely for the recent release of several data products where protein levels can be associated with genetic information for thousands of individuals. In the current work, authors have contrasted sex-differential protein abundance with sex-specific genetic regulation for 4,775 unique proteins, targeted by 4,979 unique aptamers in 4,403 females and 3,945 males (aged 29–64) from the Fenland study and 1,463 unique proteins, targeted by 1,463 unique antibody assays among 56 25,904 females and 22,113 males (aged 49-60) from UK Biobank.

The described results are very interesting and constitute a nice benchmark for future studies using the referenced technologies. Authors report that, despite a large number of proteins with significantly different mean expression levels between males and females, there is little evidence for large genetic effects at play in explaining these differences.

As a resource, the current manuscript provides several sensitivity-type analyses that deserve mention. As a suggestion, is ST2, it would be interesting if authors could add separate columns for the sensitivity analysis adjustment for each covariate instead of a column with all covariates in the same model (BMI, LDL, ALT, smoking status and alcohol consumption).

In addition, some extra sensitivity analysis would be most welcome. In particular, an analysis using only "healthy" individuals from UKBB and Fenland (in the case that selecting individuals without any medical condition proves too challenging, at least excluding individuals with cardiovascular disease and hypertension would be informative) could help in understanding whether the sex differences in the levels of NT-proBNP and troponin T are due to biological differences in cellular and protein turn-over across sexes or being confounded by different prevalences of individuals affected by ischemic heart disease, hypertension, or other factor.

An important point that is only tangentially discussed is how to accommodate the apparent paradoxical observation in which although protein levels are expressed differently between women and men, there is no evidence for an important genetic root in this phenomenon. The potential explanations given by authors don't appear to convey a unifying framework to explain the observations. Rather than a criticism, this makes the current report even more interesting and an appealing starting point for the broader discussion about the biological differences between sexes.

Several other important points deserve comment.

The very definition of a sex-specific effect is not totally discussed and, because it might prove controversial, I would like to suggest the authors address this point in the discussion paragraphs.

It is indeed not very clear from the Methods section what the authors considered to be sex-specific effects and how this was calculated. I suggest rephrasing the methods section for clarity.

In addition, it might be argued that instead of running a stratified analysis and then comparing heterogeneity results from a meta-analysis, a better approach would be to use an individual-level approach (since individual-level data was available to authors) and look for significant interaction effects between SNVs and sex. Another approach would be to also take into consideration the main effects (sex and SNV) to define sex-specific regulation (as proposed through an iterative modeling approach by groups exploring gene x environment interactions in genetic studies, PMID: 28620071 as an example). Regardless of the approach used, results should not differ significantly, but an extended discussion of these points is believed to be warranted. Why did the authors opt for a meta-analysis approach when platforms were not actually meta-analyzed and an individual-level approach could potentially offer more control over the modeling process?

Public availability of the results is important and how these results will be made available should be explicit in the manuscript. This is especially relevant if considering the entirety of the summary statistics (and not only the sd-pQTLs) for the sex-stratified protein genome-wide association analysis which would, by itself, justify the publication of this manuscript as an important resource to the community.

Another important point within the approach taken by authors refers to the way normalized relative protein expression levels are determined and the different steps both used platforms take to guarantee generalizability and mappability between different batches and different experimental runs. Although not exactly the same among the two used platforms, there are steps where experiment-wide statistics are used in the downstream data processing. It is not clear what are the consequences of these decisions in biasing the results of a sex-stratified analysis (in fact, of any stratified analysis). It would be extremely helpful if the authors could provide as supplementary methods and results a comprehensive analysis showing that reprocessing the raw data for both technologies, considering only males and females separately does not affect overall results. This would have to be done with the help of the companies responsible for both technologies, but it would be extremely valuable as added information to the immense community of researchers using these technologies.

Finally, the concordance between platforms did not appear to be particularly high. Only 47.3% of overlapping targets showed significant and directionally concordant effects for sex. Unfortunately, it was difficult to derive the specific numbers of concordance. These results should be made clearer (Supplementary Table 2 has all the information, but a Supplementary Figure with the correlation of the estimated betas for proteins represented in both platforms would be welcome). Keeping in mind that the comparison of mean values between sexes is well-powered and that 1,101 proteins were targeted by both platforms and 68% of proteins were different between sexes, one would expect a higher number of concordant results.

Minor comments:

It is appreciated that the authors acknowledge the fact that sex and gender are used interchangeably in the current work, but that these are not the same concepts.

Typo on ST2, description paragraph. druggability instead of draggability.

Line 326, remove "for"

Line 218, typo lacking a "be" in '(...) might possibly be the result (...)'

Although not something extremely problematic, it did not escape our attention the fact that models for evaluating the sex differences in protein abundances are being adjusted differently among the two studies. For a systematic evaluation of this variable, having the same model for both studies would be better. It is, however, not expected that this fact is biasing the results in any way.

No particular enrichment of sd-pQTLs was observed for proteins whose genes are located in the X chromosome. Can the authors discuss the potential effect of X inactivation in their analysis?

Reviewer #2

(Remarks to the Author)

(0) Summary

Sex differences across quantitative and disease traits are widespread across the human phenome. Differential gene expression across human tissues has been described extensively, particularly in the landmark 2020 GTEx v8 paper by Oliva et al. In addition, sex differences in genetic architecture between complex traits have been described (Bernabeu et al, 2021), as well as sex-biased/sex-differential eQTLs (sb or sd-eQTLs, Oliva et al, 2020). In recent years, proteomics has taken center stage, particularly with the 2023 release of the UK Biobank's Pharma Proteomics Project, which presents an incredible opportunity to continue unveiling the molecular mechanisms underlying sex differences at the blood protein level.

This study looks to do just that, making use of data from the Fenland study (proteome measured using the aptamer-based Somalomic platform), and data from the UK Biobank (proteome measured with the antibody-based OLINK platform). The Fenland study measures around 4.8K unique proteins across around 8.3K samples, whilst the UK Biobank measures around 1.5K unique proteins across around 50K samples. There is an overlap of around 1K proteins measured by both platforms.

The study is mainly broken down into three simple parts: (1) Search for differential protein levels between the sexes, (2) Search for sex-differential genetic regulation of protein levels (sd-pQTLs), and (3) linking the latter to sex differences in the genetic architecture of complex traits, as measured in the UK Biobank. Point (1) finds that indeed, as was observed with gene expression in the past, differences in protein levels are widespread (around 2/3s of measured proteins presenting differential levels). Point (2) finds a small number of sd-pQTLs – 3.9% and 0.3% of proteins measured in the Fenland and UKBB studies, respectively, for a total of 74 sd-pQTLs. Finally, point (3) finds that these sd-pQTLs don't translate into sex-differential disease risk.

Overall, this is an important effort that looks to fill a hole in our understanding of sex differences in human disease. However, some extra work, clarification, and finessing could help this paper be even more impactful and helpful for the field, as described in my comments below.

(1) Abstract

Upon first read of the abstract I was left confused about the data used, sample sizes, and purpose of the study. I believe this needs to be clarified heavily, and readers should know immediately what sample sizes were employed. An extra sentence giving context to why sex differences in human health are important at the beginning of the abstract could be beneficial to understand WHY the study is being performed. Also, the first sentence uses the term “sex differences” twice, I'd consider changing this. Finally, I wonder if “none of the sex-differential pQTLs translated into sex-differential disease risk” captures what was done accurately enough. Instead maybe opt to say “[...] genetically mediated sex-differential disease risk”.

(2) Sex differences vs sex dimorphism

Throughout the paper a differentiation has been made between sex differential genetic protein regulation and sex dimorphic genetic protein regulation. The line that divides the two is whether a sb-pQTL presents opposite signed effects in male and female specific pQTL models and/or whether a sb-pQTL is significant ONLY in one of the sex-specific pQTL models.

I am not convinced by this separation, and would argue that these are all different flavours of sex differences, and opt out of using the term sex dimorphism altogether, particularly when talking about sd-pQTLs that were only found to be significant in one of the sexes, as is this lack of significance could just be statistical (lack of power to detect a small effect) vs biological, which is what the name entails.

However, I realize the use of “sex differences” and “sex dimorphism” is varied in the literature, so I'd be keen so hear what the other reviewers say on this as well. I would personally talk about sex differences and group results into different categories as (1) same direction of effect, (2) opposite signed effect, and (3) only significant in one sex (though I believe the latter to be less relevant for reasons stated above).

Additionally, and importantly, this separation also minimizes what could potentially be very interesting findings: indeed, even if an sd-pQTL is significant in both sexes with the same direction of effect, the difference in magnitude of effect could still be large and provide interesting insights. An example of this is MME, shown to have a very large beta difference in Figure 2, and it's not mentioned in the main text.

(3) Differences in methods for aptamer and antibody based studies

To evaluate sex differences in protein abundance, different models are used for aptamer and antibody based measurements. For example, for Olink age2 is included in the model, but this is not the case for the Somalogic measurements. Why is this?

Similarly, different approaches are used to evaluate sex-biases in pQTLs for each platform, for example using fastGWA vs REGENIE, one platform being corrected for 10 genomics PCs and the other not. Genetic variant QC parameters are also not identical. I would urge authors to clarify why these choices were made, and discuss how they could affect results.

(4) Assessment of sex differences

In this study, sex differences in pQTL effects are established by measuring heterogeneity in a meta-analysis of female and male specific models. This is a less conventional method than what other papers in the field have done, which generally involve either (1) a statistical test explicitly assessing a difference in effect, or (2) the inclusion of a GxS interaction term in a model. I'd be keen to read why the authors chose this method in the text. I also wonder if a sensitivity analysis comparing this method to the others mentioned would be good to warrant further trust in the results.

(5) Masking analysis

When describing the sd-pQTL found for CDH15 with opposite signed effects, the authors describe how, given opposite signs, the male and female effects cancel each other out in a sex-combined analysis, thus masking a potentially interesting pQTL in sex-agnostic models. This kind of masking can also happen when QTLs have the same direction of effect or when QTLs are just significant in one of the sexes (see Bernabeu et al, 2021 Nature Gen). I wonder if it would be good to conduct a more extensive analysis to see if this masking is more widespread.

(6) Overlap with eQTLs

Potentially worth performing colocalization analysis with eQTL data or at least checking for overlap between sex-biased pQTLs found and sex-biased eQTLs.

(7) Enrichment

Are sd-pQTLs found enriched for a particular characteristic? A given gene-set?

(8) Additional limitations to discuss

Some extra additional limitations to include in the discussion include (1) effects of diversity, since only people from white European background are studied here, and (2) the fact that only a specific period of the human lifespan is represented here, mainly 40-50 year olds. Indeed, the proteome, particularly the sex-differing proteome, could vary through the human lifespan.

(9) Other comments

The current introduction in the main text is quite brief and does not provide a current state-of-the-field that provides appropriate context for the study. I am left wondering what makes this study important – is it the largest of its nature? What other studies assessing sex biased pQTLs have come before it? What does this study bring to the table that others haven't? Personally, I'd mention Somalogic and OLINK in the introduction of the main text instead of just saying aptamer- and antibody-based assays, but I'll leave this up to the authors.

When giving results, sometimes integers are given, sometimes percentages, sometimes both. I'd be consistent, and use both when appropriate.

Lines 64-65: first time Bonferroni correction is mentioned, I'd explicitly say what the thresholds were for each technology and how they were obtained for each platform (number of tests).

Lines 66-67: "...overlapping targets with significant and directionally concordant effects". – No need to say significant, this is already implied by first part of sentence, unless it means they were significant in their respective male and female specific models, in which case, clarify. In regards to "directionally concordant effects" – clarify if this is between the sexes, or between the two platforms used.

Lines 67-68: "Results exemplified large differences between the sexes, with a slightly larger number of protein targets showing higher levels in males compared to females across both technologies" – Explicitly say the numbers/percentages.

Lines 73-74: "9.34% and 18.63%, respectively" – Indicate that its respectively to the two platforms – this isn't clear in the sentence.

Lines 79-81: "while others likely reflect the effect of sex-differences in body composition on plasma abundance of specific protein targets, such as leptin or adiponectin" – Add a reference for this?

Lines 87-88: "We identified a total of 92 proteins that are the targets of already approved drugs or drugs in early clinical trials". How was this evaluated? Not mentioned in Methods either.

Lines 89-91: "While plasma protein levels are not the primary target for most of those drugs, our results can potentially help understanding sex-differential drug effects." Give an example of one of the targets found?

Figure 1: In the lower plot I'd consider using two different colors instead of two different shades of green.

Lines 124-127: Mentions of enrichment of sd-pQTLs are made, but how this is assessed is not mentioned, nor commented on in the Methods. Clarify exactly how it was evaluated. Also clarify what "clear bias" means explicitly.

Line 128: CDH15 is mentioned for the first time but full gene name is not given, unlike for the rest of genes, be consistent. Instead it is given further down in lines 143-144.

Line 142: Explicitly say which two proteins presented more than 1 sd-pQTL in the first sentence of the paragraph.

Figure 2: Is this Figure correct? Expected to see CDH15 twice given it has two sd-pQTLs, but its only present once, and the sd-pQTL which presents opposite signs in the plot is listed as PSG7.

Figure 3: I'd describe this figure in more detail, particularly the x axis of the Miami plot. Not clear it represents the 74 pQTLs x 365 traits evaluated at first glance.

Lines 185-186: From the text it is not clear that a PheWAS is being conducted, clarify, also stating how the thresholds were obtained etc.

Lines 219-220: "We obtained some evidence that larger sample sizes can identify a greater number of significant sd-pQTLs" – where in the study is this discussed/described? I didn't get this from reading the paper – if it is there do make more clear. Smaller sd-pQTL studies were not mentioned in introduction.

Line 367: "locus" instead of "loci".

No comments are made regarding what the strengths and weaknesses of these two different platforms (Olink and Somalogic) measure in any part of the main text or methods, and how they could differ and how they could potentially influence results. For the overlapping proteins, were protein levels between the two technologies compared, particularly per sex, as sanity checks?

Were any sd-pQTLs found across BOTH platforms? If not, potentially say explicitly somewhere in text.

Version 1:

Reviewer comments:

Reviewer #1

(Remarks to the Author)

The authors provide a carefully revised version of the manuscript.

The work is very timely and the results contribute new insights to an area that has been somehow overlooked during the advent of this new biomarker agnostic technologies era.

Their answers satisfy all my previous comments/suggestions. The results of this work should be highly useful to the community and will help a better understanding of the complex architecture of polygenic traits.

Minor comments:

line 236: typo "singifiacnt", to "significant"

line 244: typo "which is plays", to "which plays"

line 259: rephrase for better comprehension

line 335: I suggest modify to "In this study, we observed and replicated substantial variation ..."

Reviewer #2

(Remarks to the Author)

I thank the authors for the large undertaking of revising the manuscript as per mine and Reviewer 1's comments (which I appreciate were not few!). I believe the manuscript now reads much better, is clear, concise, and the importance and impact are well understood.

I believe the authors have addressed all my major comments. The only one I believe could still be tackled a bit more is that regarding the use of meta-analyses and heterogeneity tests vs interaction models or explicit statistical tests to assess difference in effect. I appreciate the justification the authors have provided and believe its justified for such a large number of tests, but would have liked to see a small sensitivity analysis – possibly for just one protein as a proof of concept – to see how this approach differs in comparison to more commonly seen models in the field. I however leave this up to the editor, as I believe this is a strong manuscript already.

Besides that, just a few minor comments:

1. L36: sd-pQTL abbreviation is given without an introduction. I'd have it as: Most of these 103 sexually different pQTLs (sd-pQTLs) [...]
2. L85: Two significance thresholds are given but its not mentioned why – I'd do as in Figure 1 and mention its for aptamer & antibody based studies respectively.
3. L90: "abd" is and?
4. Figure 1 legend: think authors forgot to change "darker green" for "yellow"/"orange" now that the fig has been updated.
5. L40: "identified" used twice in same sentence.
6. L142-143: "we observed that around 15% of pQTLs females" > "we observed that around 15% of pQTLs identified in females"?
7. L152-153: Think you have these flipped based on the text: should it not be 1 with opposite signed effects and 30 that were only sig in one sex?
8. L178: Should Spint3 be capitalized?
9. L189: "Differed" > "differ".
10. L189: "For example, we the cis-sd-pQTL" – Remove the "we"
11. L192: pQTL or QTL? Also, if this is also a pQTL for APOE specify, as its not super clear.
12. L196: "Hormon" > "hormone".
13. L213: "Singifiacnt" > "significant".

14. L221: Parenthesis should be after rs113693994, not before?
15. Figure 2: the figure caption references filled and hollow circles, but I'm seeing all circles as filled – did the authors mean a change in opacity to highlight significance?
16. Figure 3A: Be careful with the “typo” highlighting in the figure axis labels.
17. L295: Remove the “here”.
18. L455: In their rebuttal authors state that reason for differences in models (dif covariates etc) used for aptamer and antibody based data is they are following gold-standard proteogenomic protocols from past papers – I would potentially state this explicitly in text if it hasn't already and I missed it.
19. L456: “was” > “were”.
20. L508-511: I'd have the justification for meta-analyses as its own paragraph – currently gets lost.

REVIEWER COMMENTS

We thank you for the opportunity to revise our work that has clearly improved upon addressing the helpful comments by the reviewers. We have now addressed all comments, carefully revised the manuscript and extended our analysis to all available proteogenomic data by performing the same analyses for the additional 1,462 protein measurements. We outline our response and all changes in our point-by-point response to the reviewers below (reviewer comments have been numbered):

Reviewer #1 (Remarks to the Author):

In this manuscript, Koprulu et al. evaluate the presence of sex-specific genetic associations for circulating protein levels. This is an extremely important aspect of the contemporary approach to mapping genetic determinants of human phenotypes. It is also a fundamental aspect in the process of understanding the nature of gene expression regulation, as well as whether differences between sexes are at play and how to take them into account when conducting genetic association studies. The authors should be commended for this work.

Their work is timely for the recent release of several data products where protein levels can be associated with genetic information for thousands of individuals. In the current work, authors have contrasted sex-differential protein abundance with sex-specific genetic regulation for 4,775 unique proteins, targeted by 4,979 unique aptamers in 4,403 females and 3,945 males (aged 29–64) from the Fenland study and 1,463 unique proteins, targeted by 1,463 unique antibody assays among 25,904 females and 22,113 males (aged 49-60) from UK Biobank.

The described results are very interesting and constitute a nice benchmark for future studies using the referenced technologies. Authors report that, despite a large number of proteins with significantly different mean expression levels between males and females, there is little evidence for large genetic effects at play in explaining these differences.

We would like to thank the reviewer for their supportive evaluation of our work and helpful comments.

1. As a resource, the current manuscript provides several sensitivity-type analyses that deserve mention. As a suggestion, is ST2, it would be interesting if authors could add separate columns for the sensitivity analysis adjustment for each covariate instead of a column with all covariates in the same model (BMI, LDL, ALT, smoking status and alcohol consumption).

R1 response 1 We thank the reviewer for their useful suggestion. We have followed their suggestion and now added a separate column for the sensitivity analysis performed for each covariate to ST2 and clarified this in the methods (p17, line 457-463).

2. In addition, some extra sensitivity analysis would be most welcome. In particular, an analysis using only "healthy" individuals from UKBB and Fenland (in the case that selecting individuals without any medical condition proves too challenging, at least excluding individuals with cardiovascular disease and hypertension would be informative) could help in understanding whether the sex differences in the levels of NT-proBNP and troponin T are due to biological differences in cellular and protein turn-over across sexes or being confounded by different prevalences of individuals affected by ischemic heart disease, hypertension, or other factor.

R1 response 2 We agree with the reviewer that ideally an unselected, 'disease-free' population would be best to establish genuine sex differences, that are not explained by differential medical histories across the sexes. This was actually one advantage of Fenland study, which is a population-based cohort with participants most of whom were 'disease-free' and treatment naive at the time of baseline visit where they underwent detailed phenotyping.

Additionally, our sensitivity analysis accounting for powerful surrogates of metabolic diseases, such as BMI or LDL-cholesterol, already indicated that some but not all sex-differences in protein levels are explained by such effects. For UK Biobank, restricting UK Biobank to a 'disease-free' subset would also introduce selection biases that might also lead to spurious associations. Most importantly, higher or lower levels of a particular protein in one of the sexes may tell little about the relevance for disease risk, which would require sex-differential risk analysis. In fact, this was the reason to perform such analysis using pQTLs as genetic anchor.

We acknowledge in the revised version of the manuscript, that our approach to account for possible confounding in sex-differential non-genetic analysis is incomplete (p13, line 337-350).

3. An important point that is only tangentially discussed is how to accommodate the apparent paradoxical observation in which although protein levels are expressed differently between women and men, there is no evidence for an important genetic root in this phenomenon. The potential explanations given by authors don't appear to convey a unifying framework to explain the observations. Rather than a criticism, this makes the current report even more interesting and an appealing starting point for the broader discussion about the biological differences between sexes.

R1 response 3 We agree with the reviewer and are as staggered about this apparent discrepancy. We have now extended the discussion to provide potential explanations, including, but not limited to, i) small scale effects across the entire genome rather than individual loci that cumulatively account for sex-differences, ii) missing information on potential confounding factors, including early life exposures/lifestyle (see R1 response 2), or iii) higher order interactions of genomic loci with sex hormones that determine gene expression (p12, line 298-304; p12 line 310 – 320; p13, line 337-350).

Several other important points deserve comment.

4. The very definition of a sex-specific effect is not totally discussed and, because it might prove controversial, I would like to suggest the authors address this point in the discussion paragraphs.

R1 response 4 We followed the helpful suggestion by the reviewer and have now refined our definition of 'sex-differential' (p5, line 135 – 138; p6 line 150 -155). In the revised manuscript, we also discuss possible implications and acknowledge the limitation that 'sex-discordant' pQTLs might reach significance in yet larger studies in the opposite sex while still being characterized by substantial effect size differences (p6, line 150-155).

5. It is indeed not very clear from the Methods section what the authors considered to be sex-specific effects and how this was calculated. I suggest rephrasing the methods section for clarity.

R1 response 5 We apologize for the unclear definition, that has now been revised in the results (see R1 response 4) and methods sections (p19, line 495-501).

6. In addition, it might be argued that instead of running a stratified analysis and then comparing heterogeneity results from a meta-analysis, a better approach would be to use an individual-level approach (since individual-level data was available to authors) and look for significant interaction effects between SNVs and sex. Another approach would be to also take into consideration the main effects (sex and SNV) to define sex-specific regulation (as proposed through an iterative modeling approach by groups exploring gene x environment interactions in genetic studies, PMID: 28620071 as an example). Regardless of the approach used, results should not differ significantly, but an extended discussion of these points is believed to be warranted. Why did the authors opt for a meta-analysis approach when platforms were not actually meta-analyzed and an individual-level approach could potentially offer more control over the modeling process?

R1 response 6 We thank the reviewer for their comment. Prior to starting this project, we assessed the statistical performance and efficiency of different methods for testing for sex-differences. Running a stratified analysis and then comparing heterogeneity results from a meta-analysis approximates the individual level interaction tests yet is several fold more computationally and time-efficient than the individual level interaction tests. Given the scale of this study (over 11.2 million variants, and 3 X 7,902 unique protein targets = ~ 265 billion association tests overall), we have decided to use a meta-analyses approach and a stringent significance threshold to ensure selection of biologically relevant signals. Notably, testing pQTLs previously discovered as main effects (Pietzner et al. 2021 Science and Koprulu et al. 2023 Nature Metabolism) did not yield additional robust sd-pQTLs considering more lenient testing criteria but rather marginal, significant effect size differences for pQTLs with overall strong effect sizes.

We added a statement to the revised method section to justify our analytical approach (p19, line 509-510).

7. Public availability of the results is important and how these results will be made available should be explicit in the manuscript. This is especially relevant if considering the entirety of the summary statistics (and not only the sd-pQTLs) for the sex-stratified protein genome-wide association analysis which would, by itself, justify the publication of this manuscript as an important resource to the community.

R1 response 7 We fully agree with the reviewer regarding the importance of open access and data availability for enhancing scientific progress. We will submit all sex-stratified protein genome-wide association summary statistics to the GWAS Catalog upon acceptance of this manuscript to make the data available to the scientific community.

8. Another important point within the approach taken by authors refers to the way normalized relative protein expression levels are determined and the different steps both used platforms take

to guarantee generalizability and mappability between different batches and different experimental runs. Although not exactly the same among the two used platforms, there are steps where experiment-wide statistics are used in the downstream data processing. It is not clear what are the consequences of these decisions in biasing the results of a sex-stratified analysis (in fact, of any stratified analysis). It would be extremely helpful if the authors could provide as supplementary methods and results a comprehensive analysis showing that reprocessing the raw data for both technologies, considering only males and females separately does not affect overall results. This would have to be done with the help of the companies responsible for both technologies, but it would be extremely valuable as added information to the immense community of researchers using these technologies.

R1 response 8 The reviewer brings up an important point, that differences between strata in a population can be introduced by improper randomization during study design. In other words, we would expect to see artificial sex-differences in protein levels if sexes were not well balanced across batches. However, we did not observe significant differences of sex ratios in either the UK Biobank (Tab. 1) or Fenland (Tab. 2) study by batch or plate, a result of considering sex as a variable for randomization. We further accounted for technical variables, such as test site, that may indeed be different across the sexes in our analyses.

Notably, processing sexes separately would aggravate sex-differential analysis, since both assays report only relative units rather than absolute concentrations making it impossible to establish whether differential effect estimates for pQTL are due to actual differences in how molar concentrations change or whether effects are similar, but standard deviations differ between the sexes.

Table 1: Distribution of males and females across batches in UKBB.

Batch Number	Number of females	Number of males	% of females	% of males
P1	34	28	54.84	45.16
P2	832	708	54.03	45.97
P3	2832	2337	54.79	45.21
P4	6375	5388	54.20	45.80
P5	4042	3419	54.18	45.82
P6	4040	3533	53.35	46.65
P7	4075	3404	54.49	45.51
P8	3667	3290	52.71	47.29

Table 2: Distribution of males and females across batches in Fenland study.

Batch Number	Number of females	Number of males	% of females	% of males
P1	40	33	54.79	45.21
P2	3	2	60.00	40.00
P3	40	34	54.05	45.95
P4	31	40	43.66	56.34
P5	12	18	40.00	60.00
P6	44	31	58.67	41.33
P7	52	26	66.67	33.33
P8	37	35	51.39	48.61
P9	5	7	41.67	58.33
P10	36	40	47.37	52.63
P11	38	35	52.05	47.95

P12	34	36	48.57	51.43
P13	29	42	40.85	59.15
P14	36	34	51.43	48.57
P15	37	31	54.41	45.59
P16	25	28	47.17	52.83
P17	35	34	50.72	49.28
P18	37	36	50.68	49.32
P19	35	40	46.67	53.33
P20	26	27	49.06	50.94
P21	34	33	50.75	49.25
P22	32	33	49.23	50.77
P23	40	38	51.28	48.72
P24	5	4	55.56	44.44
P25	35	32	52.24	47.76
P26	35	35	50.00	50.00
P27	39	38	50.65	49.35
P28	29	34	46.03	53.97
P29	39	31	55.71	44.29
P30	37	29	56.06	43.94
P31	43	32	57.33	42.67
P32	35	33	51.47	48.53
P33	36	32	52.94	47.06
P34	35	39	47.30	52.70
P35	41	30	57.75	42.25
P36	7	2	77.78	22.22
P37	42	30	58.33	41.67
P38	31	35	46.97	53.03
P39	31	31	50.00	50.00
P40	41	31	56.94	43.06
P41	34	31	52.31	47.69
P42	12	13	48.00	52.00
P43	41	35	53.95	46.05
P44	27	35	43.55	56.45
P45	40	42	48.78	51.22
P46	36	26	58.06	41.94
P47	43	37	53.75	46.25
P48	38	35	52.05	47.95
P49	6	3	66.67	33.33
P50	32	39	45.07	54.93
P51	27	43	38.57	61.43
P52	43	30	58.90	41.10
P53	4	2	66.67	33.33
P54	32	42	43.24	56.76
P55	46	32	58.97	41.03
P56	36	36	50.00	50.00
P57	43	28	60.56	39.44
P58	5	2	71.43	28.57
P59	34	38	47.22	52.78
P60	17	9	65.38	34.62
P61	40	31	56.34	43.66
P62	40	27	59.70	40.30
P63	15	14	51.72	48.28
P64	38	37	50.67	49.33
P65	36	27	57.14	42.86

P66	1	1	50.00	50.00
P67	3	3	50.00	50.00
P68	35	39	47.30	52.70
P69	40	32	55.56	44.44
P70	35	44	44.30	55.70
P71	39	35	52.70	47.30
P72	38	35	52.05	47.95
P73	20	21	48.78	51.22
P74	36	26	58.06	41.94
P75	37	32	53.62	46.38
P76	39	37	51.32	48.68
P77	40	34	54.05	45.95
P78	15	17	46.88	53.13
P79	40	35	53.33	46.67
P80	9	8	52.94	47.06
P81	38	35	52.05	47.95
P82	34	44	43.59	56.41
P83	38	38	50.00	50.00
P84	38	20	65.52	34.48
P85	40	29	57.97	42.03
P86	39	29	57.35	42.65
P87	36	38	48.65	51.35
P88	41	36	53.25	46.75
P89	31	29	51.67	48.33
P90	29	34	46.03	53.97
P91	40	39	50.63	49.37
P92	44	30	59.46	40.54
P93	23	41	35.94	64.06
P94	4	3	57.14	42.86
P95	3	4	42.86	57.14
P96	43	27	61.43	38.57
P97	34	38	47.22	52.78
P98	36	27	57.14	42.86
P99	20	9	68.97	31.03
P100	35	37	48.61	51.39
P101	41	25	62.12	37.88
P102	44	31	58.67	41.33
P103	44	29	60.27	39.73
P104	40	33	54.79	45.21
P105	36	37	49.32	50.68
P106	38	29	56.72	43.28
P107	41	32	56.16	43.84
P108	4	6	40.00	60.00
P109	36	34	51.43	48.57
P110	42	33	56.00	44.00
P111	32	42	43.24	56.76
P112	22	20	52.38	47.62
P113	33	38	46.48	53.52
P114	44	31	58.67	41.33
P115	42	34	55.26	44.74
P116	37	27	57.81	42.19
P117	33	29	53.23	46.77
P118	34	33	50.75	49.25
P119	46	30	60.53	39.47

P120	45	24	65.22	34.78
P121	30	26	53.57	46.43
P122	0	2	0.00	100.00
P123	18	17	51.43	48.57
P124	40	34	54.05	45.95
P125	37	37	50.00	50.00
P126	38	34	52.78	47.22
P127	48	29	62.34	37.66
P128	43	28	60.56	39.44
P129	26	34	43.33	56.67
P130	35	37	48.61	51.39
P131	28	40	41.18	58.82
P132	26	18	59.09	40.91
P133	36	36	50.00	50.00
P134	37	35	51.39	48.61
P135	48	24	66.67	33.33
P136	45	30	60.00	40.00

9. Finally, the concordance between platforms did not appear to be particularly high. Only 47.3% of overlapping targets showed significant and directionally concordant effects for sex. Unfortunately, it was difficult to derive the specific numbers of concordance. These results should be made clearer (Supplementary Table 2 has all the information, but a Supplementary Figure with the correlation of the estimated betas for proteins represented in both platforms would be welcome). Keeping in mind that the comparison of mean values between sexes is well-powered and that 1,101 proteins were targeted by both platforms and 68% of proteins were different between sexes, one would expect a higher number of concordant results.

R1 response 9 We appreciate the importance to provide a clear representation of cross-platform consistencies, since those protein targets will also have the highest prior of representing biologically most relevant associations. With the inclusion of the entire UKB protein data set, we now have a total of 1,991 overlapping protein targets (UniProt ID) of which 40.9% (n=815) showed significant and directionally consistent differences across the sexes. We have now added a more detailed overview as a Supplementary Figure (Supplementary Figure 1). We observed an overall correlation of $r=0.60$ that we have now added in the figure legend of this Supplementary Figure 1.

As we (Pietzner et al., Nat Comms, 2021) and others (Katz et al., Science Advances, 2022; Eldajrn et al., Nature, 2023) previously demonstrated, inconsistencies between platforms could be due to a wide range of reasons including but not limited to: (i) differences in power between the different cohorts, (ii) differences in cohort-related participant and sample characteristics, (iii) technological differences between the platforms (e.g. targeting different isoforms, differing epitope effects on the assay binding). In line with this, we replicated previous observations (Pietzner et al., Nat Comms, 2021, Katz et al., Science Advances, 2022; Eldajrn et al., Nature, 2023), that protein targets that are poorly correlated protein targets (based on correlations calculated by Eldajrn et al., Nature, 2023) did also show discordance of observational results, including differences in plasma levels between the sexes. The mean correlation for overlapping protein targets which were significant and directionally consistent in both had a mean correlation coefficient (r) of 0.43 (SD=0.27), whereas the overlapping protein targets which were significant in both but directionally inconsistent or were significant in only one technology had a correlation coefficient of 0.19 (SD=0.27) and 0.26 (SD=0.29), respectively. We have now added these findings to the Supplementary Figure 1, however, we agree that these are

important factors to consider and future benchmarking studies or orthogonal technologies would be needed to resolve some of the disparities.

Supplementary Figure 1: Comparison of effect size estimates of sex for overlapping protein targets between the measurements from antibody-based and aptamer-based proteomics technologies.

There were 1,991 unique protein target combinations between the two platforms targeting 1,838 unique proteins (defined by unique UniProt ID). Of these, findings for 1,006 were significant in both technologies (815 of which were also directionally consistent), 770 were significant in only one technology and 215 were not significant in any of these technologies. Overall, the correlation between the effect size estimates from the two technologies was $r=0.60$ for overlapping targets. The mean correlation (based on correlations by Eldjarn et al., Nature, 2023 (1)) for overlapping protein targets which were significant and directionally consistent in both technologies had a mean correlation coefficient (r) of 0.43 ($SD=0.27$), whereas the overlapping protein targets which were significant in both technologies but directionally inconsistent or were significant in only one technology had a correlation coefficient of 0.19 ($SD=0.27$) and 0.26 ($SD=0.29$), respectively.

Minor comments:

10. It is appreciated that the authors acknowledge the fact that sex and gender are used interchangeably in the current work, but that these are not the same concepts.

R1 response 10 We apologize for lack of clarity, but we do not refer to gender in the entire manuscript when reporting the results of our study, since the available data did not allow us to specify the gender participants self-identify with. We clarified our approach in the introduction (page 3, line 73-80).

11. Typo on ST2, description paragraph. Druggability instead of draggability.

R1 response 11 We thank the reviewer and have now corrected this typo.

12. Line 326, remove "for"

R1 response 12 We thank the reviewer and have now revised this sentence.

13. Line 218, typo lacking a "be" in '(...) might possibly be the result (...)'

R1 response 13 We thank the reviewer and have now revised this sentence.

14. Although not something extremely problematic, it did not escape our attention the fact that models for evaluating the sex differences in protein abundances are being adjusted differently among the two studies. For a systematic evaluation of this variable, having the same model for both studies would be better. It is, however, not expected that this fact is biasing the results in any way.

R1 response 14 We thank the reviewer for their comment. To explain our workflow, we have first started this project by analysing the proteomics data in Fenland study using fastGWA. In the meantime, the UK Biobank data has become available and computationally more efficient association software for genome-wide association studies have become available such as REGENIE. We sanity checked for 100 protein targets that both softwares have provided almost identical association results. In terms of similar yet slightly different variables being adjusted for the two cohorts, this was due to differences in the cohort characteristics, and we have followed the recommendations of the flagship efforts from these cohorts (Pietzner et al., Science, 2021; Sun et al., Nature., 2023).

15. No particular enrichment of sd-pQTLs was observed for proteins whose genes are located in the X chromosome. Can the authors discuss the potential effect of X inactivation in their analysis?

R1 response 15 We agree with the reviewer that X-chromosome inactivation can potentially mask the effects of true biological relevance of certain X-chromosome variants and their sex-differential impact on the protein abundance. However, given the scale of analyses conducted in this study (over 11.2 million variants, and 3 X 7,902 unique protein targets = ~ 265.5 billion association tests overall), we could not identify computationally efficient, scalable X-chromosome test software. However, we have tested the X-chromosome variants by keeping the default variant calls for females (i.e. 0,1,2) and classifying males as homozygous wild-type or alternate allele carrier (i.e. 0,2). However, we now acknowledge the limitation that we were not able to systematically consider the impact of X-chromosome inactivation in this study (p13, line 337-350).

Reviewer #2 (Remarks to the Author):

0. Summary

Sex differences across quantitative and disease traits are widespread across the human phenome. Differential gene expression across human tissues has been described extensively, particularly in the landmark 2020 GTEx v8 paper by Oliva et al. In addition, sex differences in genetic architecture between complex traits have been described (Bernabeu et al, 2021), as well as sex-biased/sex-differential eQTLs (sb or sd-eQTLs, Oliva et al, 2020). In recent years, proteomics has taken center stage, particularly with the 2023 release of the UK Biobank's Pharma Proteomics Project, which presents an incredible opportunity to continue unveiling the molecular mechanisms underlying sex differences at the blood protein level.

This study looks to do just that, making use of data from the Fenland study (proteome measured using the aptamer-based Somalogic platform), and data from the UK Biobank (proteome measured with the antibody-based OLINK platform). The Fenland study measures around 4.8K unique proteins across around 8.3K samples, whilst the UK Biobank measures around 1.5K unique proteins across around 50K samples. There is an overlap of around 1K proteins measured by both platforms.

The study is mainly broken down into three simple parts: (1) Search for differential protein levels between the sexes, (2) Search for sex-differential genetic regulation of protein levels (sd-pQTLs), and (3) linking the latter to sex differences in the genetic architecture of complex traits, as measured in the UK Biobank. Point (1) finds that indeed, as was observed with gene expression in the past, differences in protein levels are widespread (around 2/3s of measured proteins presenting differential levels). Point (2) finds a small number of sd-pQTLs – 3.9% and 0.3% of proteins measured in the Fenland and UKBB studies, respectively, for a total of 74 sd-pQTLs. Finally, point (3) finds that these sd-pQTLs don't translate into sex-differential disease risk.

Overall, this is an important effort that looks to fill a hole in our understanding of sex differences in human disease. However, some extra work, clarification, and finessing could help this paper be even more impactful and helpful for the field, as described in my comments below.

1. Abstract

Upon first read of the abstract I was left confused about the data used, sample sizes, and purpose of the study. I believe this needs to be clarified heavily, and readers should know immediately what sample sizes were employed. An extra sentence giving context to why sex differences in human health are important at the beginning of the abstract could be beneficial to understand WHY the study is being performed. Also, the first sentence uses the term "sex differences" twice, I'd consider changing this. Finally, I wonder if "none of the sex-differential pQTLs translated into sex-differential disease risk" captures what was done accurately enough. Instead maybe opt to say "[...] genetically mediated sex-differential disease risk".

R2 response 1 We followed the helpful suggestions of the reviewer and have now rephrased the abstract accordingly (page 2, line 26-43), including a clear rationale for and conclusion of the study.

2. Sex differences vs sex dimorphism

Throughout the paper a differentiation has been made between sex differential genetic protein regulation and sex dimorphic genetic protein regulation. The line that divides the two is whether a sb-pQTL presents opposite signed effects in male and female specific pQTL models and/or whether a sb-pQTL is significant ONLY in one of the sex-specific pQTL models.

I am not convinced by this separation, and would argue that these are all different flavours of sex differences, and opt out of using the term sex dimorphism altogether, particularly when talking about sd-pQTLs that were only found to be significant in one of the sexes, as is this lack of significance could just be statistical (lack of power to detect a small effect) vs biological, which is what the name entails.

However, I realize the use of “sex differences” and “sex dimorphism” is varied in the literature, so I’d be keen so hear what the other reviewers say on this as well. I would personally talk about sex differences and group results into different categories as (1) same direction of effect, (2) opposite signed effect, and (3) only significant in one sex (though I believe the latter to be less relevant for reasons stated above). Additionally, and importantly, this separation also minimizes what could potentially be very interesting findings: indeed, even if an sd-pQTL is significant in both sexes with the same direction of effect, the difference in magnitude of effect could still be large and provide interesting insights. An example of this is MME, shown to have a very large beta difference in Figure 2, and it’s not mentioned in the main text.

R2 response 2 We agree with the reviewer, that ‘sex differences’ and ‘sex dimorphism’ are fragile concepts and have now adopted a definition in line with the reviewer’s suggestion (p6, line 150-158). We also thank the reviewer for pointing out errors with Figure 2 (SPINK1 was wrongly labelled as MME), that has now been corrected. The large sex difference observed for sd-pQTL for SPINK1 is likely due to its minor allele frequency, which is at the lowest boundary of MAF threshold included in this study (MAF>1%). As the reviewer emphasizes, we discussed all major differences in effect sizes of sd-pQTL, since they may well have different downstream consequences.

Now within the updated manuscript

3. Differences in methods for aptamer and antibody based studies

To evaluate sex differences in protein abundance, different models are used for aptamer and antibody based measurements. For example, for Olink age2 is included in the model, but this is not the case for the Somalogic measurements. Why is this?

Similarly, different approaches are used to evaluate sex-biases in pQTLs for each platform, for example using fastGWA vs REGENIE, one platform being corrected for 10 genomics PCs and the other not. Genetic variant QC parameters are also not identical. I would urge authors to clarify why these choices were made, and discuss how they could affect results.

R2 response 3 We now provide a detailed justification for analytical approaches, that are outlined in detail in **R1 response 14**. In terms of covariates, we thank the reviewer for pointing this out. We have now revised the methods to clarify the covariates included for each study. When selecting the covariates, we have followed the covariates the flagship proteogenomic efforts each study (Pietzner et al. 2021, Sun et al. 2023) have used.

4. Assessment of sex differences

In this study, sex differences in pQTL effects are established by measuring heterogeneity in a meta-analysis of female and male specific models. This is a less conventional method than what other papers in the field have done, which generally involve either (1) a statistical test explicitly assessing a difference in effect, or (2) the inclusion of a GxS interaction term in a model. I'd be keen to read why the authors chose this method in the text. I also wonder if a sensitivity analysis comparing this method to the others mentioned would be good to warrant further trust in the results.

R1 response 4 We thank the reviewer for their comment. Prior to embarking on this project, we have assessed the performance and efficiency of different methods to assess sex-differences. Running a stratified analysis and then comparing heterogeneity results from a meta-analysis approximates the individual level interaction tests yet is several folds more computationally and time-efficient than the individual level interaction tests. Given the scale of this study (over 11.2 million variants, and 3 X 7,902 unique protein targets = ~ 47 billion association tests overall), we have decided to use a meta-analyses approach and use a stringent significance threshold to ensure selection of biologically relevant signals.

5. Masking analysis

When describing the sd-pQTL found for CDH15 with opposite signed effects, the authors describe how, given opposite signs, the male and female effects cancel each other out in a sex-combined analysis, thus masking a potentially interesting pQTL in sex-agnostic models. This kind of masking can also happen when QTLs have the same direction of effect or when QTLs are just significant in one of the sexes (see Bernabeu et al, 2021 Nature Gen). I wonder if it would be good to conduct a more extensive analysis to see if this masking is more widespread.

R2 response 5 We thank the reviewer for their useful suggestion. We have now systematically assessed the masking of all pQTLs identified in each sex. We identified that around 15% of pQTLs females (n=1149/7424) and males (n=976/6546) for aptamer-based technology and around 7% of pQTLs identified in females (n=1332/18307) or males (n=995/14305) for antibody-based technology were not significant ($p < 5 \times 10^{-8}$) reported in the sex-combined analyses, as their effect were masked in the sex-combined analyses, which has now been included in the revised manuscript (p5, line 141-145).

6. Overlap with eQTLs

Potentially worth performing colocalization analysis with eQTL data or at least checking for overlap between sex-biased pQTLs found and sex-biased eQTLs.

R2 response 6 We apologize if this information was not provided in sufficient detail in the previous version of the manuscript. We systematically tested for an overlap between sex-differential pQTLs we identified with the sex-biased eQTLs reported in Oliva et al., Science (2020). However, we only observed one finding (rs8942 with C4BPB with stronger in females in GTEx and stronger in males in our study). One explanation for the missing overlap might be the small sample size in GTEx compared to 20-100x larger pQTL studies to identify effects across strata. Additionally, complex regulatory processes beyond transcriptional regulation can account for the limited convergence or event divergent findings between eQTLs and pQTLs in general (Pietzner et al. Science 2020, Sun et al. Nature 2023).

7. Enrichment

Are sd-pQTLs found enriched for a particular characteristic? A given gene-set?

R2 response 7 We thank the reviewer for their suggestion. We have conducted pathway enrichment analyses for the protein targets that we observed at least one sd-pQTL for in each technology, however, did not observe a significant enrichment for any pathway.

8. Additional limitations to discuss

Some extra additional limitations to include in the discussion include (1) effects of diversity, since only people from white European background are studied here, and (2) the fact that only a specific period of the human lifespan is represented here, mainly 40-50 year olds. Indeed, the proteome, particularly the sex-differing proteome, could vary through the human lifespan.

R2 response 8 We fully agree with the reviewer and have now added these points to our discussion (p13, line 333-336).

Other comments

9. The current introduction in the main text is quite brief and does not provide a current state-of-the-field that provides appropriate context for the study. I am left wondering what makes this study important – is it the largest of its nature? What other studies assessing sex biased pQTLs have come before it? What does this study bring to the table that others haven't?

R2 response 9 We thank the reviewer for their suggestion, we have now expanded our introduction to put our study into context (p3, line 51-62). In summary, although the extent of sex-differences have been systematically assessed in relation to eQTLs (Oliva et al. Science, 2020), this is the first systematic assessment of sex-differential pQTLs to date. Previous studies (Pietzner et al., Science 2021, Koprulu et al., 2023) have assessed the extent of sex-differences among pQTLs identified in sex-combined analysis, however, sex-agnostic models can mask the impact of sex-differential pQTLs in sex-combined analyses. The large enough sample sizes in Fenland Study (4,403 females, 3,945 males) and UK Biobank (25,904 females, 22,113 males) and broad proteomic coverage has now allowed us to systematically assess the sex-differential proteome and extent to which genetic factors contribute to these differences.

10. Personally, I'd mention Somalogic and OLINK in the introduction of the main text instead of just saying aptamer- and antibody-based assays, but I'll leave this up to the authors.

R2 response 10 We agree that specifications of the assay vendors would make presentation of the results more accurate, but we deliberately decided not to mention vendors to minimize any perception as a promotion work. However, we now mention them in the introduction (p3, line 68-72).

11. When giving results, sometimes integers are given, sometimes percentages, sometimes both. I'd be consistent, and use both when appropriate.

R2 response 11 We have now revised this and always include both to be consistent.

12. Lines 64-65: first time Bonferroni correction is mentioned, I'd explicitly say what the thresholds were for each technology and how they were obtained for each platform (number of tests).

R2 response 12 We have now revised this sentence to clarify (p4, line 85).

13. Lines 66-67: "...overlapping targets with significant and directionally concordant effects". – No need to say significant, this is already implied by first part of sentence, unless it means they were significant in their respective male and female specific models, in which case, clarify. In regards to

“directionally concordant effects” – clarify if this is between the sexes, or between the two platforms used.

R1 response 13 We thank the reviewer and have now revised this sentence to clarify that we mean directionally concordant effects between sexes.

14. Lines 67-68: “Results exemplified large differences between the sexes, with a slightly larger number of protein targets showing higher levels in males compared to females across both technologies” – Explicitly say the numbers/percentages.

R1 response 14 We thank the reviewer and have now revised this sentence to clarify (p4, line 88-90).

15. Lines 73-74: “9.34% and 18.63%, respectively” – Indicate that its respectively to the two platforms – this isn’t clear in the sentence.

R1 response 15 We have now revised this sentence to clarify that its respectively to two sensitivity analyses (p4 line 94-96).

16. Lines 79-81: “while others likely reflect the effect of sex-differences in body composition on plasma abundance of specific protein targets, such as leptin or adiponectin” – Add a reference for this?

We thank the reviewers, we have now added references that demonstrate the association between leptin and adiponectin and body fat distribution (p4, 101-106).

17. Lines 87-88: “We identified a total of 92 proteins that are the targets of already approved drugs or drugs in early clinical trials”. How was this evaluated? Not mentioned in Methods either.

R2 response 17 In addition to the reference, we have now clarified this in the methods (page 4 line 109; page 18 line 463-465). Briefly, we mapped protein coding genes to drug targets based on the work by Finan et al. Science Translational Medicine 2017.

18. Lines 89-91: “While plasma protein levels are not the primary target for most of those drugs, our results can potentially help understanding sex-differential drug effects.” Give an example of one of the targets found?

R2 response 18 We have now revised the results to include an example for this statement, Tenecteplase or Urokinase which have been described to be differentially effective in female and male patients in post stroke therapy (Reeves et al. 2008) (page 4, line 111-114).

19. Figure 1: In the lower plot I’d consider using two different colors instead of two different shades of green.

R2 response 19 We thank the reviewer and have now revised Figure 1 accordingly.

20. Lines 124-127: Mentions of enrichment of sd-pQTLs are made, but how this is assessed is not mentioned, nor commented on in the Methods. Clarify exactly how it was evaluated. Also clarify what “clear bias” means explicitly.

R2 response 20 We apologize for the missing information and have now revised the method section to contain more information about the downstream analyses or lookups we have done for the sd-pQTLs, including specific enrichment tests. (p9, line 512-517).

21. Line 128: CDH15 is mentioned for the first time but full gene name is not given, unlike for the rest of genes, be consistent. Instead it is given further down in lines 143-144.

R2 response 21 We thank the reviewer for spotting this inconsistency and corrected it in the revised version of the manuscript accordingly.

22. Line 142: Explicitly say which two proteins presented more than 1 sd-pQTL in the first sentence of the paragraph.

R2 response 22 We rephrased the corresponding sentences following the advice of the reviewer.

23. Figure 2: Is this Figure correct? Expected to see CDH15 twice given it has two sd-pQTLs, but its only present once, and the sd-pQTL which presents opposite signs in the plot is listed as PSG7.

R2 response 23 We thank the reviewer for pointing out errors that had happened when plotting and labelling results from sd-pQTL analysis where the results were presented correctly in Supplementary Tables 3 and 4, but are shown incorrectly in the previous version of Figure 2. We have now revised this figure to contain correct information and also added the additional sd-pQTLs discovered through the analysis of additional protein targets in UK Biobank.

24. Figure 3: I'd describe this figure in more detail, particularly the x axis of the Miami plot. Not clear it represents the 74 pQTLs x 365 traits evaluated at first glance.

R2 response 24 We followed the advice of the reviewer and provide now more details in the corresponding figure legend.

25. Lines 185-186: From the text it is not clear that a PheWAS is being conducted, clarify, also stating how the thresholds were obtained etc.

R2 response 25 We have now revised the text and added the relevant information (p11, line 270-284).

26. Lines 219-220: "We obtained some evidence that larger sample sizes can identify a greater number of significant sd-pQTLs" – where in the study is this discussed/described? I didn't get this from reading the paper – if it is there do make more clear. Smaller sd-pQTL studies were not mentioned in introduction.

R2 response 26 We apologize for the missing context and rephrased the section accordingly (pX, line YYY). Briefly, this argument was mainly driven when comparing results from Fenland (n~8k) to UK Biobank (n~40k).

27. Line 367: "locus" instead of "loci".

R2 response 27 We thank the reviewer and have now revised this sentence.

28. No comments are made regarding what the strengths and weaknesses of these two different platforms (Olink and Somalogic) measure in any part of the main text or methods, and how they

could differ and how they could potentially influence results. For the overlapping proteins, were protein levels between the two technologies compared, particularly per sex, as sanity checks?

R2 response 28 We appreciate the importance of technological differences and now include a detailed presentation in Supplementary Figure 1 and its legend and discussion (p13, line 345-349) on how this may have affected results. Briefly, we replicated previous observations, that protein targets that are poorly correlated (Pietzner et al., Nat Comms, 2021, Katz et al., Science Advances, 2022; Eldajrn et al., Nature, 2023) did also show discordance of observational results, including differences in plasma levels between the sexes (see R1 response 9). Unfortunately, we were not able to systematically test, whether correlations between platforms differed by sex.

29. Were any sd-pQTLs found across BOTH platforms? If not, potentially say explicitly somewhere in text.

R2 response 29 For a total of 53 out of 92 unique protein targets (defined by UniProt ID, for which there was at least one sd-pQTL identified) there was protein targets were measured on both platforms, but we identified consistent sd-pQTL evidence for 4 protein targets, in line with poorly correlating assay results. This might be best explained by poor assay performance masking sd-pQTLs, but also less statistical power to detect more subtle differences in the Fenland cohort. We observed consistent evidence for sd-pQTLs only for NCAM-1, PZP, NEU1 and ENPP7 and now highlight these in the main text (p6, line 167-168 & 178-182). We also added an extended section on cross-platform (in)consistencies to the main text.

REVIEWERS' COMMENTS

We thank the reviewers for their positive overview about the importance of our work. We have now carefully addressed the helpful comments by the reviewers.

Reviewer #1 (Remarks to the Author):

The authors provide a carefully revised version of the manuscript.

The work is very timely and the results contribute new insights to an area that has been somehow overlooked during the advent of this new biomarker agnostic technologies era.

Their answers satisfy all my previous comments/suggestions. The results of this work should be highly useful to the community and will help a better understanding of the complex architecture of polygenic traits.

Minor comments:

1. line 236: typo "singificant", to "significant"

R1 response 1 We thank the reviewer for spotting the typo that has now been revised.

2. line 244: typo "which is plays", to "which plays"

R1 response 2 We thank the reviewer for spotting the typo that has now been revised.

3. line 259: rephrase for better comprehension

R1 response 3 We followed the suggestion and rephrased the sentence accordingly.

4. line 335: I suggest modify to "In this study, we observed and replicated substantial variation ..."

R1 response 4 We followed the help suggestion of the reviewer and revised accordingly.

Reviewer #2 (Remarks to the Author):

I thank the authors for the large undertaking of revising the manuscript as per mine and Reviewer 1's comments (which I appreciate were not few!). I believe the manuscript now reads much better, is clear, concise, and the importance and impact are well understood.

1. I believe the authors have addressed all my major comments. The only one I believe could still be tackled a bit more is that regarding the use of meta-analyses and heterogeneity tests vs interaction models or explicit statistical tests to assess difference in effect. I appreciate the justification the authors have provided and believe its justified for such a large number of tests, but would have liked to see a small sensitivity analysis – possibly for just one protein as a proof of concept – to see how this approach differs in comparison to more commonly seen models in the field. I however leave this up to the editor, as I believe this is a strong manuscript already.

R2 response 1 We understand and share concerns about the validity of heterogeneity tests compared to proper interaction testing. We have now run a sensitivity analysis for all sd-pQTLs identified in UK Biobank and observed a strong correlation ($r=0.98$) between the heterogeneity p-value estimated from the meta-analyses and p-value observed from the G*S interaction term (model run: inverse rank normalized protein abundance \sim SNP + sex + SNP*sex + PC1 + PC2 + PC3 + PC4 + PC5 + PC6 + PC7 + PC8 + PC9 + PC10 + age + age2 + Batch + chip). The comparison of p-values obtained from both approaches can be seen in the figure below. We have now revised the manuscript and added the results of this sensitivity analysis, demonstrating good coherence between both modelling approaches. The notably somewhat stronger p-values in the interaction model are likely due to loss of precision in meta-analysis but did not change any conclusions of the paper as they only occur why beyond corrected statistical significance.

Figure: Comparison of the heterogeneity p-value estimated from the meta-analyses and p-value observed from the G*S interaction term for significant sd-pQTLs in UK Biobank (n=88)

Besides that, just a few minor comments:

2. L36: sd-pQTL abbreviation is given without an introduction. I'd have it as: Most of these 103 sexually different pQTLs (sd-pQTLs) [...]

R2 response 2 We thank the reviewer for pointing this out, we have now revised the abstract.

3. L85: Two significance thresholds are given but its not mentioned why – I'd do as in Figure 1 and mention its for aptamer & antibody based studies respectively.

R2 response 3 We thank the reviewer for pointing this discrepancy out, that has now been revised in the updated manuscript.

4. L90: "abd" is and?

R2 response 4 This has now been revised.

5. Figure 1 legend: think authors forgot to change "darker green" for "yellow"/"orange" now that the fig has been updated.

R2 response 5 We thank the reviewer for pointing this out, we have now revised the Figure 1 legend.

6. L40: “identified” used twice in same sentence.

R2 response 6 We thank the reviewer, and this has now been revised.

7. L142-143: “we observed that around 15% of pQTLs females” > “we observed that around 15% of pQTLs identified in females”?

R2 response 7 We thank the reviewer, and this has now been revised.

8. L152-153: Think you have these flipped based on the text: should it not be 1 with opposite signed effects and 30 that were only sig in one sex?

R2 response 8 We thank the reviewer for pointing this out, this has now been corrected.

9. L178: Should Spint3 be capitalized?

R2 response 9 We have now revised the text to indicate the results refer to the mouse orthologue.

10. L189: “Differed” > “differ”.

R2 response 10 We thank the reviewer, and this has now been revised.

11. L189: “For example, we the cis-sd-pQTL” – Remove the “we”

R2 response 11 We thank the reviewer, and this has now been revised.

12. L192: pQTL or QTL? Also, if this is also a pQTL for APOE specify, as its not super clear.

R2 response 12 We thank the reviewer, and this has now been revised.

13. L196: “Hormon” > “hormone”.

R2 response 13 We thank the reviewer, and this has now been revised.

14. L213: “Singifiactn” > “significant”.

R2 response 14 We thank the reviewer, and this has now been revised.

15. L221: Parenthesis should be after rs113693994, not before?

R2 response 15 We thank the reviewer, to avoid overcomplicating the sentence we have given the rsID of the sd-pQTL in the parentheses.

16. Figure 2: the figure caption references filled and hollow circles, but I’m seeing all circles as filled – did the authors mean a change in opacity to highlight significance?

R2 response 16 We thank the reviewer for their comment. Some of the circles in the bottom “sex-discordant pQTLs” panel are hollow for males or females. However, 95% confidence interval bar still runs through them. We have now increased the resolution when we export the figure and hope that this will improve the visualization.

17. Figure 3A: Be careful with the “typo” highlighting in the figure axis labels.

R2 response 17 We thank the reviewer, and this has now been revised.

18. L295: Remove the “here”.

R2 response 18 We thank the reviewer, and this has now been revised.

19. L455: In their rebuttal authors state that reason for differences in models (dif covariates etc) used for aptamer and antibody based data is they are following gold-standard proteogenomic protocols from past papers – I would potentially state this explicitly in text if it hasn’t already and I missed it.

R2 response 19 We now state this by clearly citing the relevant references.

20. L456: “was” > “were”.

R2 response 20 We thank the reviewer, and this has now been revised.

21. L508-511: I’d have the justification for meta-analyses as its own paragraph – currently gets lost.

R2 response 20 We thank the reviewer, and this has now been revised and the results of sensitivity analyses have been added (see R2 response 1).